# Make-An-Agent: A Generalizable Policy Network Generator with Behavior-Prompted Diffusion

**Yongyuan Liang**[12]    **Tingqiang Xu**[3]    **Kaizhe Hu**[3]    **Guangqi Jiang**[4]
**Furong Huang**[2]    **Huazhe Xu**[13]

[1] Shanghai Qi Zhi Institute    [2] University of Maryland, College Park
[3] Tsinghua University    [4] University of California, San Diego

## Abstract

Can we generate a control policy for an agent using just one demonstration of desired behaviors as a prompt, as effortlessly as creating an image from a textual description? In this paper, we present **Make-An-Agent**, a novel policy parameter generator that leverages the power of conditional diffusion models for behavior-to-policy generation. Guided by behavior embeddings that encode trajectory information, our policy generator synthesizes latent parameter representations, which can then be decoded into policy networks. Trained on policy network checkpoints and their corresponding trajectories, our generation model demonstrates remarkable versatility and scalability on multiple tasks and has a strong generalization ability on unseen tasks to output well-performed policies with only few-shot demonstrations as inputs. We showcase its efficacy and efficiency on various domains and tasks, including varying objectives, behaviors, and even across different robot manipulators. Beyond simulation, we directly deploy policies generated by **Make-An-Agent** onto real-world robots on locomotion tasks.

## 1   Introduction

Policy learning traditionally involves using sampled trajectories from a replay buffer or behavior demonstrations to learn policies or trajectory models mapping from state $s$ to action $a$, modeling a narrow behavior distribution. In this paper, we consider a shift in paradigm: moving beyond training a policy, can we reversely predict optimal policy network parameters using suboptimal trajectories from offline data? This approach would obviate the need to explicitly model behavior distributions, allowing us to learn the underlying parameter distributions in the parameter space, thus revealing the implicit relationship between agent behaviors for specific tasks and policy parameters.

Using low-dimensional demonstrations (such as agent behavior) to guide the generation of high-dimensional outputs (policy parameters) is a challenging problem. When diffusion models [12, 20] have demonstrated highly competitive performance on various tasks including text-to-image synthesis, we are inspired to approach policy network generation as a conditional denoising diffusion process. By progressively refining noise into structured parameters, the diffusion-based generator can discover various policies that are not only superior in performance but also more robust and efficient than the demonstration in the policy parameter space.

While prior works on hypernetworks [10, 1, 18] explore the concept of training a hypernetwork to generate weights for another neural network, they primarily use hypernetworks as an initialization network of meta-learning [7] and then adapt to specific task settings. Our approach diverges from this paradigm by leveraging agent behaviors as direct prompts or to generate optimal policies within the

---

[i]Code, dataset and video are released in https://cheryyunl.github.io/make-an-agent/.
 Corresponding to: cheryunl@umd.edu

38th Conference on Neural Information Processing Systems (NeurIPS 2024).

parameter space, without the need for any downstream policy fine-tuning or adaptation with gradient updates. Since behaviors - as the observable manifestation of deployed policies - from different tasks often share underlying skills or environmental information, our policy generator can exploit these potential correlations in the parameter space, such as shared parameters for similar motion patterns, which leads to enhanced cross-task one-shot generalizability. What we need is an end-to-end behavior-to-policy generator, not a shared base policy.

To achieve this, we introduce **Make-An-Agent**, featuring three key technical contributions: (1) We propose an autoencoder that encodes policy networks into compact latent representations based on their network layers, which can also effectively reconstruct the original policy from its latent representation. (2) We leverage contrastive learning to capture the mutual information between long-term trajectories and their success or future states. This approach yields a novel and efficient behavior embedding. (3) We utilize a simple yet effective diffusion model conditioned on the learned behavior embeddings, to generate policy parameter representations, which are then decoded into deployable policies using the pretrained decoder. (4) We construct a pretrained dataset of policy network parameters and corresponding deployed trajectories to train our proposed methodology.

To investigate the generation performance of **Make-An-Agent**, we evaluate our approach in three continuous control domains including diverse tabletop manipulation and real-world locomotion tasks. During test time, we generate policies using trajectories from the replay buffer of partially-trained RL agents. The policies generated by our method demonstrate superior performance compared to policies produced by multi-task [28, 23] or meta learning [7, 27] and other hypernetwork-based generation methods [1]. Our generator offers several key advantages:

- **Versatility**: **Make-An-Agent** excels in generating effective policies for a wide range of tasks by conditioning on agent behavior embeddings. Since we train the parameter generator for latent parameter representations, it can generate policy networks of varying sizes within the latent space, demonstrating scalability.

- **Generalizability:** Our diffusion-based generator demonstrates robust generalization, yielding proficient policies even for unseen behaviors or unseen embodiments in unfamiliar tasks.

- **Robustness**: Our method can generate diverse policy parameters, exhibiting resilient performance under environmental randomness from simulators and real-world environments. Notably, **Make-An-Agent** can synthesize high-performing policies when fed with noisy trajectories, highlighting the robustness of our model.

## 2  Backgrounds

**Policy Learning.**  Reinforcement Learning (RL) is structured within the formation of Markov Decision Processes (MDPs) [2], which is defined by the tuple $M = \langle \mathcal{S}, \mathcal{A}, P, \mathcal{R}, \gamma \rangle$. Here, $\mathcal{S}$ signifies the state space, $\mathcal{A}$ the action space, $P$ the transition probabilities, $\mathcal{R}$ the reward function and $\gamma$ the discount factor. RL aims to optimize an agent's policy $\pi : \mathcal{S} \rightarrow \mathcal{A}$, which outputs action $a_t$ based on state $s_t$ at each timestep, to maximize cumulative rewards. The optimal policy can be expressed as:

$$\pi^* = \arg\max_{\pi} \mathbb{E}_{z \sim \pi} \left[ \sum_{t=0}^{\infty} \gamma^t r_t \right], \tag{1}$$

where $z$ represents a trajectory generated by following policy $\pi$. In deep RL, policies $\pi$ are represented using neural network function approximations [22], parameterized by $\theta_\pi$, facilitating the learning of intricate behaviors across high-dimensional state and action spaces.

**Diffusion Models.**  Denoising Diffusion Probabilistic Models (DDPMs) [12] are generative models that frame data generation through a structured diffusion process, which involves iteratively adding noise to the data and then denoising it to recover the original signal. Given a sample $x_0$, the forward diffusion process to obtain $x_1, x_2, ..., x_T$ of increasing noise intensity is typically denoted by:

$$q(x_t \mid x_{t-1}) = \mathcal{N}(x_t, \sqrt{1 - \beta_t} x_{t-1}, \beta_t I), \tag{2}$$

where $q$ is the forward process, $\mathcal{N}$ is Gaussian noise, and $\beta_t \in (0, 1)$ is is the noise variance.

The denoising process, which is the reverse of the forward diffusion, can be formulated as:

$$p_\theta(x_{t-1} \mid x_t) = \mathcal{N}(x_{t-1} \mid \mu_\theta(x_t, t), \Sigma_\theta), \tag{3}$$

where $p_\theta$ denotes the reverse process, $\mu_\theta$ and $\Sigma_\theta$ are the mean and variance of the Gaussian distribution respectively, which can be approximated by a noise prediction neural network parameterized by $\theta$.

Diffusion models aim to learn reverse transitions that maximize the likelihood of the forward transitions at each time step $t$. The noise prediction network $\theta$ is optimized using the following objective, as the function mapping from $\epsilon_\theta(x_t, t)$ to $\mu_\theta(x_t, t)$ is a closed-form expression:

$$\mathcal{L}_{\text{DM}}(\theta) := \mathbb{E}_{x_0 \sim q, \epsilon \sim \mathcal{N}(0,1), t}[||\epsilon - \epsilon_\theta(\sqrt{\bar{\alpha}_t} x_0 + \sqrt{1 - \bar{\alpha}_t} \epsilon, t)||^2], \tag{4}$$

Here, $\epsilon \sim \mathcal{N}(\mathbf{0}, \mathbf{I})$, is the target Gaussian noise, $\bar{\alpha}_t := \prod_{s=1}^{t} 1 - \beta_s$, and $\sqrt{\bar{\alpha}_t} x_0 + \sqrt{1 - \bar{\alpha}_t} \epsilon$ is the estimated distribution of $x_t$ from the closed-form relation.

Although diffusion models are typically used for image generation through the reverse process, the variable $x$ can be generalized to represent diverse entities for generation. In this paper, we adapt $x$ to represent the parameters $\theta_\pi$ of the policy network in policy learning.

## 3 Methodology

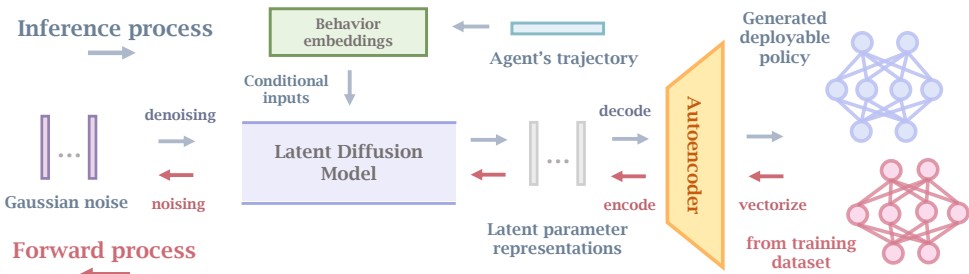

Figure 1: **Overview:** In **the inference process** of policy parameter generation, conditioning on behavior embeddings from the agent's trajectory, the latent diffusion model denoises random noise into a latent parameter representation, which can then be reconstructed as a deployable policy using the autoencoder. **The forward process** for progressively noising the data is also conducted on the latent space after encoding policy parameters as latent representations.

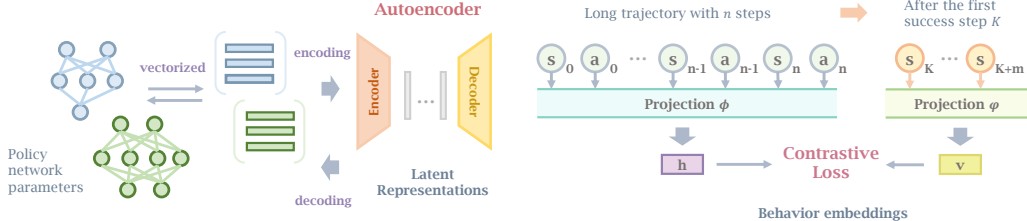

Figure 2: **Autoencoder:** Encoding policy parameters into the latent space and decoding latent parameter representations into policy networks.

Figure 3: **Contrastive behavior embeddings:** Learning informative behavior embeddings from long trajectories with contrastive loss.

**Overview.** An overview of our proposed methodology is illustrated in Figure 1. To achieve this, we address several key challenges: (1) Developing latent representations of high-dimensional policy parameters that can be effectively reconstructed into well-functioned policies. (2) Learning an embedding of behavior demonstrations that serves as an effective diffusion condition. (3) Training a conditional diffusion model specifically for policy parameter generation.

**Parameter representation.** We use an MLP with $m$ layers as the common policy approximator. Consequently, when the full parameters of a policy are flattened, they form a high-dimensional vector.

To enable generation with limited computational resources while retaining efficacy, and to support the generation of policies for different domains with varying state and action dimensions, we compress the policy network parameters into a latent space.

Based on the policy network architecture, we unfold the parameters following the architecture of the policy network, represented as $x = [x_0, x_1, \ldots, x_{m-1}]$, where $x_i$ denotes the flattened parameters from each layer. The encoder $\mathcal{E}$ encodes each $x_i$ as $z_i$, resulting in a parameter latent representation denoted as $z = [z_0, z_1, \ldots, z_{m-1}]$, where each $z_i$ in the latent space has the same dimension, while the decoder $\mathcal{D}$ can decode $z$ into $x$. To improve the robustness of this procedure, we introduce random noise augmentation in both encoding and decoding during training. Given each vectorized parameter as $x$, we minimize the objective as,

$$\mathcal{L} = \text{MSE}(x, \mathcal{D}(\mathcal{E}(x + \xi_{\mathcal{D}}) + \xi_{\mathcal{E}})), \tag{5}$$

where $z = \mathcal{E}(x + \xi_{\mathcal{D}})$, and $\xi_{\mathcal{E}}$ and $\xi_{\mathcal{D}}$ represent the augmented noise. The architecture of the autoencoder is shown in Figure 2.

For each domain, the autoencoder for parameter representation only needs to be trained once before parameter generation, which can handle policy parameters from different tasks. To facilitate the generalizability of the policy generator across domains, we design the latent parameter representations to have the same dimensions for different domains.

**Behavior embedding.** Since our goal in learning behavior embeddings is not to model the distribution of states and actions, but to provide conditional information for policy parameter generation, we aim for them to encapsulate both crucial environmental dynamics and the key information of the task goal. The principle behind our behavior embeddings is to learn the mutual information between preceding $n$ step trajectories and subsequent states with success signals.

$$\mathbb{I} = \mathcal{I}(s_{success}; \{s_i, a_i\}_{i=0}^n) \tag{6}$$

We propose a novel contrastive method to train behavior embeddings. In Figure 3, we present a design demonstration of our contrastive loss. For a long trajectory $\tau$, we decouple it as the $n$ initial state-action pairs $\tau^n = (s_0, a_0, s_1, a_1, \ldots, s_n, a_n)$ and the $m$ states after the first success time $K$ as $\hat{\tau} = (s_K, s_{K+1}, \ldots, s_{K+m})$. Given a batch of trajectory sequences $\{\tau_i\}_{i=1}^N$ which can be presented as $\{\tau_i^n, \hat{\tau}_i\}_{i=1}^N$, we optimize the contrastive objective [16, 31] as:

$$\mathcal{L}(\phi_\theta, \psi_\theta, W) = -\frac{1}{N} \sum_{i=1}^N \log \frac{h_i^\top W v_i}{\sum_{j=1}^N h_i^\top W v_j} \tag{7}$$

where $h_i = \phi_\theta(\tau_i^n)$ and $v_i = \psi_\theta(\hat{\tau}_i)$ are embeddings from different parts of the long trajectory $\tau_i$ and $W$ is a learnable metric that measures the similarity between embeddings $h_i$ and $v_i$.

For each trajectory $\tau$, we obtain a set of embeddings $\tau_e = \{h_i, v_i\}$. In practice, the choice of specific embeddings can be tailored to the characteristics of different tasks and trajectories. We use $(h_i, v_i)$ as the conditional input in our experiments.

**Flexibility.** With the consideration that in many scenarios, rewards are often sparse or non-existent, whereas success signals serve as a more direct indicator of whether a policy has achieved its objective. We therefore use original trajectories that exclude reward information but include success information.

For tasks without explicit success signals, such as locomotion, we segment long trajectories into multiple shorter trajectories. For each segment, we use the last $m$ states as $\hat{\tau}$ and the $0 - n$ state-action pairs as $\tau^n$. The informative behavior embeddings of a long trajectory are concatenated from the embeddings of all the trajectory segments.

This embedding approach strives to capture the essential information for generating behavior-specific policy parameters, including environmental dynamics and task goals, using the most concise representation possible from long trajectories and prioritizing flexibility and efficiency.

**Conditional policy generator.** After training the parameter autoencoder and behavior embeddings, for policy parameter $x$ and the corresponding trajectory $g$ deployed by policy $x$, we can transfer $x$ as latent parameter representation $z$ with the autoencoder $\mathcal{E}$ and trajectory $\tau$ as behavior embedding $\tau_e$. The conditional diffusion generator is trained on latent representation $z$, conditioning on $\tau_e$. We optimize the conditional latent diffusion model via the following loss function:

$$\mathcal{L}_{\text{LDM}}(\theta) := \mathbb{E}_{z, \epsilon \sim \mathcal{N}(0,1), t} \left[ \|\epsilon - \epsilon_\theta(z_t, \tau_e, t)\|_2^2 \right], \tag{8}$$

where the neural backbone $\epsilon_\theta(z_t, \tau_e, t)$ is implemented as a 1D convolutional UNet [19] parameterized by $\theta$ and t is uniformly sampled from $\{1, \ldots, T\}$. The outputs of our parameter generator can be encoded by $\mathcal{D}$ as deployable policies. During training the diffusion model, both the parameter autoencoder and behavior embedding layers are frozen, which ensures the training stability and efficiency.

**Dataset.**  We build a dataset containing tens of thousands of policy parameters and trajectories from deploying these policies. The dataset is obtained from multiple RL training across a range of tasks. We utilized the dataset to train both the autoencoder and behavior embedding models. Then we use the encoded parameter representations and behavior embeddings derived from the collected trajectory to train the conditional diffusion model for policy parameter generation.

# 4  Experiments

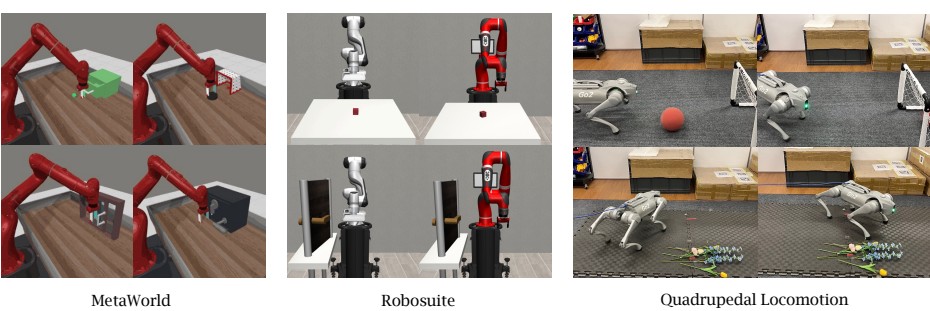

MetaWorld                    Robosuite                    Quadrupedal Locomotion

Figure 4: Visualization of MetaWorld, Robosuite, and real quadrupedal locomotion.

We conduct extensive experiments to evaluate **Make-An-Agent** , answering the following problems:

- How does our method compare with other multi-task or learning-to-learn approaches for policy learning, in terms of performance on seen tasks and generalization to unseen tasks?
- How scalable is our method, and can it be fine-tuned across different domains?
- Does our method merely memorize policy parameters and trajectories of each task, or can it generate diverse and new behaviors?

**Benchmarks.**  We include two manipulation benchmarks for simulated experiments and real-world robot tasks to show the performance and capabilities of our method as visualized in Figure 4.

**MetaWorld.**  MetaWorld [29] is a benchmark suite for robotic tabletop manipulation, consisting of a diverse set of motion patterns for the Sawyer robotic arm and interactions with different objects. We selected 10 tasks for training as **seen** tasks and 8 for evaluation as **unseen** downstream tasks. Detailed descriptions of these tasks can be found in Appendix C.1. The state space of MetaWorld consists of 39 dimensions and the action space has 4 dimensions. The policy network architecture used for MetaWorld is a 4-layer MLP with 128 hidden units, containing a total of 22,664 parameters.

**Robosuite.**  Robosuite [33], a simulation benchmark designed for robotic manipulation, supports various robots such as the Sawyer and Panda arms. We train models on three manipulation tasks: Block Lifting, Door Opening and Nut Assembly, using the single-arm Panda robot. Evaluations are conducted on the same tasks using the Sawyer robot. This experimental design aims to validate the practicality of our approach by assessing whether the generated policy can be effectively utilized on different robots. In the Robosuite environment, the state space comprises 41 dimensions, and the action space consists of 8 dimensions. The policy network employed for this domain contains 23,952 parameters.

**Quadrupedal locomotion.**  To evaluate the policies generated by **Make-An-Agent** in the real world, we utilize walk-these-ways [14] to train policies on IsaacGym and use our method to generate actor networks conditioning on trajectories from IsaacGym simulation with the pretrained adaptation modules. Then, we deploy the generated policy on real robots in environments differ from simulations. The policies generated for real-world locomotion deployment comprise 50,956 parameters.

**Dataset.** We collect 1500 policy networks for each task in MetaWorld and Robosuite. These networks are sourced from policy checkpoints during SAC [11] training. The checkpoints are saved every

5000 training steps once the test success rate reaches 1. During the training stage, we fix the initial locations of objects and goals and train the policies using different random seeds. For each task, we require an average of 8 SAC training runs, approximately 30 GPU hours.

For evaluation, the trajectories used as generation conditions are sampled from the SAC training buffer within the first 0.5 million timesteps, which can be highly sub-optimal, under the same environmental initialization. During testing, the generated policies are evaluated in 5 random initial configurations, thoroughly assessing the robustness of policies generated using trajectories from the fixed environment settings.

In RoboSuite experiments, due to the inconsistency in policy networks, we retrain the autoencoder and finetune the diffusion generator trained on MetaWorld data. The experimental setup for RoboSuite is almost identical to that of MetaWorld, with the only difference being the robot used during testing.

For real-world locomotion tasks, we save 10,000 policy network checkpoints using walk-these-ways (WTW) [14] trained on IsaacGym, requiring a total of 200 GPU hours. The 100 trajectories used as generation conditions are sourced from the first 10,000 training iterations of WTW.

**Baselines.** We compare **Make-An-Agent** with four baselines, including multi-task imitation learning (IL), multi-task reinforcement learning (RL), meta-RL with hypernetworks, and meta-IL with transformers. These represent state-of-the-art methods for multi-task policy learning and adaptation. For a fair comparison, each baseline uses the same testing trajectory data for downstream adaptation.

**Multi-task BC** [24]: We train a multi-task behavior cloning policy using trajectories in our training dataset, and then finetune it with test trajectories to adapt in specific tasks.

**Multi-task RL, CARE** [23]: We train an mixture of encoders for 2 million steps (for each task). For RL training, we train the algorithm in dense reward environments and finetune the model using test trajectories with sparse rewards, where feedback is only provided at the end of a trajectory.

**Meta-RL with hypernetworks** [1]: We train a hypernetwork with our training data with dense rewards, which can adapt to different task-specific policies during testing with test trajectories.

**Meta Imitation Learning with decision transformer(DT) [4]** We train the pre-trained DT model using the training trajectories in our dataset, then use the test trajectories from replay to adapt it to test tasks.

### 4.1 Performance Analysis

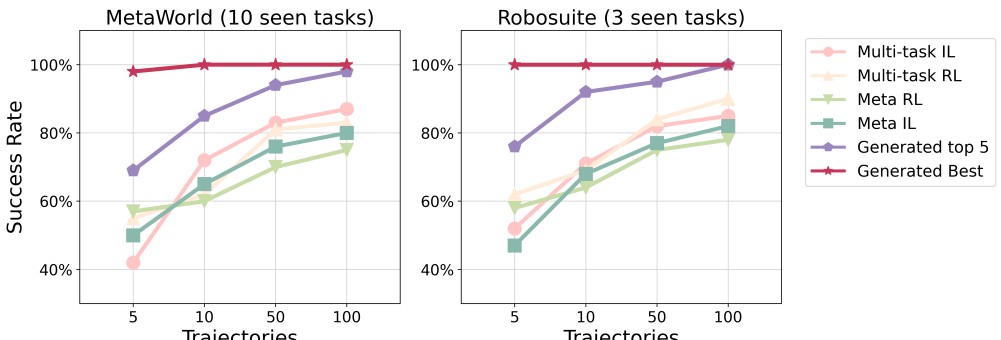

Figure 5: **Evaluation of seen tasks with 5 random initializations on MetaWorld and Robosuite.** Our method generate policies using 5/10/50/100 test trajectories. Baselines are finetuned/adapted by the same test trajectories. Results are averaged over training with 4 seeds.

By using test trajectories as conditions, our policy generator can produce an equivalent number of policy parameters. Compared with baselines, we report both the best result among the generated policies and the average performance of the top 5 policies. All algorithms use the same task-specific replay trajectories. The difference is that we use them as generation conditions, whereas other methods use them for adaptation.

We define policies achieving a 100% success rate during evaluation as qualified policies. The analysis of qualification rates for policies generated by our model is presented in Appendix C.2.

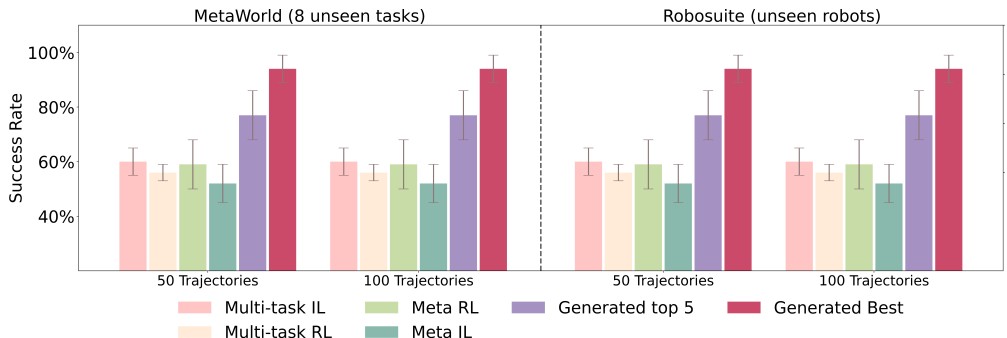

Figure 6: **Evaluation of 8 unseen tasks with 5 random initializations on MetaWorld and Robosuite.** Our method generates policies using 50/100 test trajectories without any finetuning. Baselines are adapted using the same test trajectories. Average results are from training with 4 seeds.

**Adaptability to environmental randomness on seen tasks.** Figure 5 demonstrates the significant advantage of our algorithm over other methods on seen tasks. This is attributed to the fact that, despite test trajectories originating from the same environment initialization, the generated policy parameters are more diverse, thus possessing a strong ability to adapt to environmental randomness. In contrast, other algorithms, when adapted using such singular trajectories, exhibit more limited adaptability in these scenarios. Our experimental design aligns with practical requirements, as real-world randomness is inherently more complex.

**Generalizability to unseen tasks.** Figure 6 showcases the superior performance of our algorithm on unseen tasks. Test trajectories originate from the same environment setting for each task, while evaluation occurs in randomly initialized environments. Our policy generator, without fine-tuning, directly utilizes test trajectories as input, demonstrating a remarkable ability to generate parameters that work on unseen tasks. The agent's behavior in unseen tasks exhibits similarities to seen task behaviors, such as arm dynamics and the path to goals. By effectively combining parameter representations related to these features, the generative model successfully generates effective policies. In contrast, baseline methods struggle to adapt in environmental randomness.

These results strongly suggest that our algorithm, compared to other policy adaptation methods, may offer a superior solution for unseen scenarios. To further investigate robustness in generalization, we added Gaussian noise with a standard deviation of 0.1 to actions in test trajectories used for policy generation or adaptation on unseen tasks. Figure 7 demonstrates that our method remains resilient to noisy inputs, while the performance of the baselines is significantly impacted. We believe this is because our behavior embeddings only need to capture key dynamic information as conditions to generate policies, without directly learning state-action relationships from trajectories, resulting in better robustness.

**Trajectory difference.** To compare the difference between using test trajectories as conditions and the trajectories obtained by deploying the generated policies, we visualize the trajectories during unseen task evaluations. As shown in Figure 9, our diffusion generator can synthesize various policies, which is significantly different from policy learning methods that learn to predict actions or states from trajectory data. We believe that this phenomenon fully illustrates the value of our proposed policy parameter generation paradigm.

**Parameter distinction.** Beyond trajectory differences, we also investigate the distinction between synthesized parameters and RL policy parameters. We calculate the cosine similarity between the RL policies used to obtain the test trajectories and the parameters generated from these trajectories. As a benchmark, we include the RL policies after 100 steps of finetuning with the test data. For tasks seen during training, the parameters generated by our approach demonstrate significantly greater diversity compared to the RL parameters after fine-tuning, indicating that our generator does not simply memorize training data. On unseen tasks, the similarity between our generated parameters and those learned by RL is almost negligible, with most similarities falling below 0.2. This further highlights the diversity and novelty of the policy parameters generated by our method.

**Real-world Evaluation** We further deploy policies generated from simulation trajectories onto a quadruped robot, instructing it to complete tasks as illustrated in Figure 10. Our synthesized policies

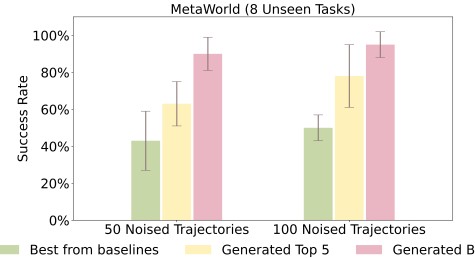
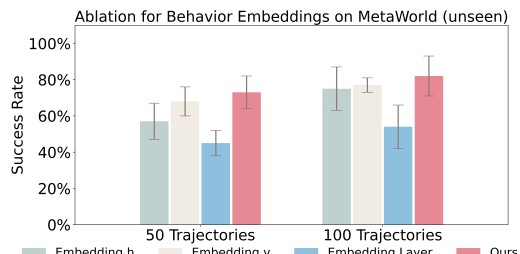

Figure 7: Evaluation of unseen tasks on Meta-World using **noised trajectories**.

Figure 8: **Ablation studies** about using different embeddings as conditions in policy generation on MetaWorld 5 unseen tasks. (Top 5 models)

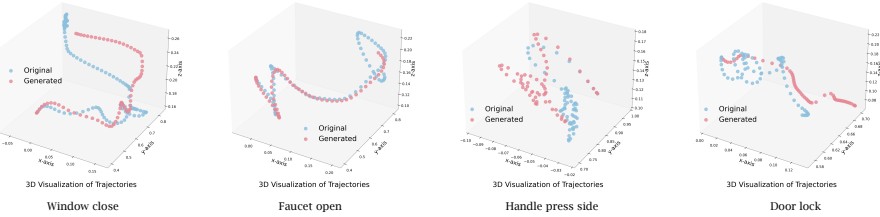

Figure 9: **Trajectory difference**: trajectories as conditional inputs v.s. trajectories from synthesized policies as outputs on MetaWorld 4 unseen tasks.

exhibit smooth and effective responses when faced with these challenging tasks, which highlights the stability of the generated policies under the dynamics randomness of real-world environments.[ii]

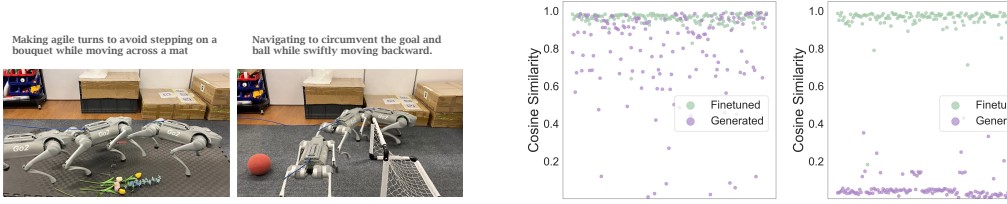

Figure 10: **Real-world locomotion tasks**, including turning, fast backward movement, and obstacle avoidance on a mat.

Figure 11: **Parameter Similarity:** Parameter cosine similarity between RL-trained policies and our generated policies or fine-tuned policies.

## 4.2 Ablation Studies

To better investigate the impact of each design choice in our method on the final results, we conduct a series of comprehensive ablation studies. All ablation studies report average results of the Top 5 generation models on MetaWorld.

**Choice of behavior embeddings.** Regarding the choice of conditional embeddings, as illustrated in Figure 3, we concatenate $h$ and $v$ as generation conditions to maximally preserve trajectory information. Figure 8 shows that utilizing either embedding individually also achieves comparable performance due to our contrastive loss, ensuring efficient capture of dynamics information. Our contrastive behavior embeddings significantly outperform a baseline that adds an embedding layer in the diffusion model to encode trajectories as input. These ablation results underscore the effectiveness of our behavior embeddings.

**Choice of trajectory length.** The trajectory length $n$ used in behavior embeddings can also impact experimental results. Figure 12a demonstrates that overly short trajectories lead to performance degradation, probably due to the absence of crucial behavior information. However, beyond 40 steps, trajectory length minimally impacts policy generation, indicating that our method is not sensitive to the length of trajectories.

---

[ii]We thank Kun Lei and Qingwei Ben for their help and support in real-robot applications.

**Impact of policy network size.** The impact of policy network size on generated parameters is also worth discussing, as the network's hidden size influences the dimensionality of parameters to be generated. Figure 12b suggests that a hidden size of 128 is a suitable choice. Smaller networks may hinder policy performance, while larger ones increase parameter reconstruction complexity.

**Impact of parameter number used in training.** We study the impact of the number of policy checkpoints included per task in the training dataset, as shown in Figure 12c. Insufficient training data (<=1000) leads to a significant performance decline across all tasks. With more than 1000 parameters, there is no notable improvement in performance.

**Impact of latent representation size.** Additionally, Figure 12d illustrates the impact of varying the size of the latent parameter representation. Larger latent representations can negatively affect the performance of the generative model. Conversely, when the size of parameter representations is too small, it may hinder the autoencoder's capacity to decode representations to deployable policies. This underscores the influence of the parameter autoencoder on the overall effectiveness of the policy network generator.

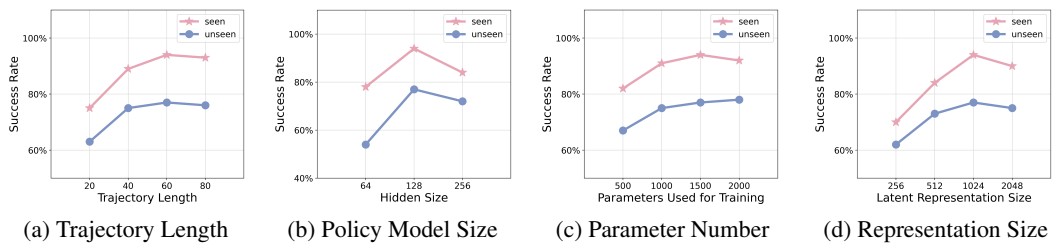

(a) Trajectory Length  (b) Policy Model Size  (c) Parameter Number  (d) Representation Size

Figure 12: Ablation studies of our technical designs on MetaWorld with 50 test trajectories (Top 5 models).

## 5 Related Works

**Parameter Generation.** Learning to generate neural networks has long been a compelling area of research. Since the introduction of Hypernetworks [10] and the subsequent extensions [3], several studies have explored neural network weight prediction. Hypertransformer [32] utilizes Transformers to generate weights for each layer of convolutional neural networks (CNN) using task samples for supervised and semi-supervised learning. Additionally, previous work [21] employs self-supervised learning to learn hyper representations of neural network weights. In the context of using diffusion models for parameter generation, G.pt [17] trains a diffusion transformer to generate parameters conditioned on learning metrics such as test losses and prediction errors, enabling the optimization of unseen parameters with a single update. Similarly, p-diff [25] propose a diffusion-based method to generate the last two normalization layers without any conditions for classification tasks. In contrast to these prior works, our focus is on policy learning problems. We develop a latent diffusion parameter generator that is more generalizable and scalable, based on agents' behaviors as prompts.

**Learning to Learn for Policy Learning.** When discussing learning to learn in policy learning, the concept of meta-learning [7] has been widely explored. The goal of meta-RL [7, 6, 9, 15] is to learn a policy that can adapt to any new task from a given task distribution. During the meta-training or meta-testing process, prior meta-RL methods require rewards as supervision for policy adaptation. Meta-imitation learning [8, 5, 27] addresses a similar problem but assumes the availability of expert demonstrations. Diffusion models have also been used in meta-learning. Metadiff [30] models the gradient descent process for task-specific adaptation as a diffusion process to propose a diffusion-based meta-learning method. Our work departs from these learning-to-learn works. Instead, we shift the focus away from data distributions across tasks and simply leverage behavior embeddings as conditional inputs for policy synthesis in the parameter space.

## 6 Conclusion

In this paper, we introduced a novel policy generation method based on conditional diffusion models. Targeting the generation of policies in high-dimensional parameter spaces, we employ an autoencoder to encode and reconstruct parameters, incorporating a contrastive loss to learn efficient behavior

embeddings. By prompting with these behavior embeddings, our policy generator can effectively produce diverse and well-performing policies. Extensive empirical results across various domains demonstrate the versatility of our approach in multi-task settings, the generalization ability on unseen tasks, and the resilience to environmental randomness. Our work not only introduces a fresh perspective on policy learning, but also establishes a new paradigm that delves into the latent connections between agent behaviors and policy parameters.

**Limitation.**  Due to the vast number of parameters involved, we have not yet explored larger and more diverse policy networks. Additionally, the capabilities of the parameter diffusion generator are limited by the parameter autoencoder. We believe there is substantial room for future research to explore more flexible parameter generation methods. It would also be interesting to apply our proposed generation framework to generate other structures, further facilitating exploration in policy learning within the parameter space.

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

# A  Broader Impact

The broader impact of our research on generating policy parameters using diffusion models is a complex interplay of potential benefits and risks. On the one hand, by enhancing the sample efficiency of policy learning in robotics, our method could accelerate the development and deployment of robotic systems across various sectors, from manufacturing and healthcare to agriculture. This could lead to increased automation, potentially improving efficiency, safety, and accessibility in these fields, ultimately enhancing the quality of life and productivity for society. Additionally, our work introduces the robotics community to the potential of policy learning in a new and promising paradigm—which could inspire further research, leading to more advanced and efficient algorithms for robot learning and control.

However, it's important to acknowledge the potential negative consequences of increased automation. Job displacement due to automation is a significant concern, as robots could replace human workers in various industries, leading to economic disruption and social unrest. Furthermore, as robots become more autonomous and capable, concerns about safety and ethical implications arise. Ensuring the safe and responsible use of robots will be paramount as our research progresses.

# B  Implementation Details

All model training are conducted on `NVIDIA A40` GPUs.

**Autoencoder.**  The autoencoder implementation consists of a three-layer MLP encoder and a decoder. Prior to training, each layer of the policy network is flattened and encoded separately. The final mean and std layers are concatenated with the middle layer for encoding.

The hyperparameters used for the autoencoder are detailed in Table 1. On average, training an autoencoder requires 5 GPU hours.

**Behavior embedding.**

The behavior embedding model consists of two three-layer MLP embeddings. During training, we concatenate the state and action sequences from the first $n = 60$ steps (each sequence having a length of 3) to form the input for the $h$ embedding layer. Subsequently, we concatenate the $m = 3$ states after success as inputs for the $v$ embedding layer. Both embedding layers output 128-dimensional vectors. When utilizing these embeddings as conditional inputs, we concatenate the $h$ and $v$ embeddings as 256-dimenional conditions.

All hyperparameters about the training of the behavior embeddings can be found in Table 2. A single training for the embeddings requires less than 1 GPU hour.

**Conditional diffusion generator.**  Our diffusion model employs a 1D convolutional U-Net architecture as its backbone, utilizing behavior embeddings as global conditions. It outputs latent parameter representations with the same dimensionality as the autoencoder's output.

Training a single diffusion generator requires only 4 GPU hours. All relevant hyperparameters are detailed in Table 3.

**Hyperparameters**  We conduct all experiments with this single set of hyperparameters.

**Data collection.**  Our dataset is collected with [13, 26] using RTX 3090 Ti.

# C  Experiments

## C.1  Task Description

**MetaWorld**  Descriptions of tasks and random initialization:

**Seen** tasks (Training):

- window open: Push and open a window. Randomize window positions
- door open: Open a door with a revolving joint. Randomize door positions
- drawer open: Open a drawer. Randomize drawer positions

Table 1: Hyperparameters for Autoencoder

| Hyper-parameter | Value |
|---|---|
| backbone | MLP |
| input dim | 22664 |
| hidden size | 1024 |
| output dim | [2, 1024] |
| encoder depth | 1 |
| decoder depth | 1 |
| input noise factor | 0.0001 |
| output noise factor | 0.001 |
| batch size | 32 |
| optimizer | AdamW |
| learning rate | 1e-3 |
| weight decay | 5e-3 |
| training epoch | 3000 |
| lr scheduler | CosineAnnealingWarmRestarts |

Table 2: Hyperparameters for Behavior Embedding

| Hyper-parameter | Value |
|---|---|
| backbone | MLP |
| trajectory dim | 1020 |
| success state dim | 117 |
| hidden size | 1024 |
| output dim | 128 |
| batch size | 16 |
| optimizer | AdamW |
| learning rate | 1e-4 |
| weight decay | 1e-4 |
| training epoch | 300 |
| lr scheduler | CosineAnnealingWarmRestarts |

Table 3: Hyperparameters for Diffusion Model

| Hyper-parameter | Value | Hyper-parameter | Value |
|---|---|---|---|
| behavior embedding shape | [256] | kernel size | 3 |
| parameter shape | [2, 1024] | noise scheduler | DDIM |
| num inference steps | 10 | batch size | 128 |
| embedding dim in diffusion steps | 128 | optimizer | AdamW |
| learning rate | 2.0e-4 | beta | [0.95, 0.999] |
| eps | 1.0e-8 | weight decay | 5.0e-4 |
| training epoch | 1000 | lr_scheduler | cosine |
| lr warmup steps | 500 | | |

- dial turn: Rotate a dial 180 degrees. Randomize dial positions
- faucet close: Rotate the faucet clockwise. Randomize faucet positions
- button press: Press a button. Randomize button positions
- door unlock: Unlock the door by rotating the lock clockwise. Randomize door positions
- handle press: Press a handle down. Randomize the handle positions
- plate slide: Slide a plate into a cabinet. Randomize the plate and cabinet positions
- reach: reach a goal position. Randomize the goal positions

**Unseen** tasks (Downstream):

- window close: Push and close a window. Randomize window positions
- door close: Close a door with a revolving joint. Randomize door positions
- drawer close: Open a drawer. Randomize drawer positions
- faucet open: Rotate the faucet counter-clockwise. Randomize faucet positions
- button press wall: Bypass a wall and press a button. Randomize the button positions
- door lock: Lock the door by rotating the lock clockwise. Randomize door positions
- handle press side: Press a handle down sideways. Randomize the handle positions
- coffee-button: Push a button on the coffee machine. Randomize the position of the button
- reach wall: Bypass a wall and reach a goal. Randomize goal positions

**Robosuite**  Descriptions of tasks, robots, and random initialization:

Tasks:

- Door: A door with a handle is mounted in free space in front of a single robot arm. The robot arm must learn to turn the handle and open the door. The door location is randomized at the beginning of each episode.

- Lift: A cube is placed on the tabletop in front of a single robot arm. The robot arm must lift the cube above a certain height. The cube location is randomized at the beginning of each episode.

- Nut Assembly - Single: Two colored pegs (one square and one round) are mounted on the tabletop, and two colored nuts (one square and one round) are placed on the table in front of a single robot arm. The goal is to place either one round nut or one square nut into its peg.

Robots:

- Panda: Panda is a 7-DoF and relatively new robot model produced by Franka Emika, and boasts high positional accuracy and repeatability. The default gripper for this robot is the PandaGripper, a parallel-jaw gripper equipped with two small finger pads, that comes shipped with the robot arm.

- Sawyer: Sawyer is Rethink Robotic's 7-DoF single-arm robot. Sawyer's default RethinkGripper model is a parallel-jaw gripper with long fingers and useful for grasping a variety of objects.

## C.2   More Results

In addition to reporting the average performance of the top 5 generated results in the main paper, we rigorously define "qualified policies" as those achieving a 100% success rate in the test environment. Table 4 presents the proportion of qualified policies among 100 policy parameters generated from 100 trajectories. Notably, we maintain an average qualification rate of over 30% on seen tasks.

Furthermore, even on unseen tasks, we can generate high-performing policies using an average of only 20 trajectories. Considering that our method does not rely on expert demonstrations, the quality and success rate of our generated policies significantly enhance the sample efficiency of policy learning.

Table 4: Qualified rate and success rate of Top 5/10 models from the generated polices with 100 trajectories on MetaWorld

| Seen Tasks | window open | door open | drawer open | dial turn | plate slide | button press | handle press | faucet close |
|---|---|---|---|---|---|---|---|---|
| Qualified Rate | $0.33 \pm 0.02$ | $0.27 \pm 0.04$ | $0.42 \pm 0.03$ | $0.23 \pm 0.02$ | $0.45 \pm 0.04$ | $0.32 \pm 0.14$ | $0.5 \pm 0.08$ | $0.45 \pm 0.13$ |
| Generated Top 5 | $1.0 \pm 0.0$ | $1.0 \pm 0.0$ | $1.0 \pm 0.0$ | $0.82 \pm 0.09$ | $1.0 \pm 0.0$ | $0.87 \pm 0.10$ | $1.0 \pm 0.0$ | $0.98 \pm 0.01$ |
| Generated Top 10 | $1.0 \pm 0.0$ | $0.87 \pm 0.06$ | $1.0 \pm 0.0$ | $0.73 \pm 0.06$ | $0.94 \pm 0.05$ | $0.80 \pm 0.05$ | $0.96 \pm 0.02$ | $0.97 \pm 0.01$ |

| Unseen Tasks | drawer close | faucet open | button press wall | coffee button | handle press side | reach wall | door lock | window close |
|---|---|---|---|---|---|---|---|---|
| Qualified Rate | $0.55 \pm 0.04$ | $0.16 \pm 0.06$ | $0.11 \pm 0.01$ | $0.08 \pm 0.03$ | $0.04 \pm 0.03$ | $0.13 \pm 0.05$ | $0.13 \pm 0.04$ | $0.10 \pm 0.01$ |
| Generated Top 5 | $1.0 \pm 0$ | $1.0 \pm 0$ | $0.97 \pm 0.02$ | $0.74 \pm 0.21$ | $0.45 \pm 0.17$ | $0.94 \pm 0.03$ | $1.0 \pm 0.0$ | $0.92 \pm 0.05$ |
| Generated Top 10 | $1.0 \pm 0$ | $0.85 \pm 0.13$ | $0.95 \pm 0.02$ | $0.64 \pm 0.23$ | $0.30 \pm 0.12$ | $0.72 \pm 0.11$ | $0.85 \pm 0.05$ | $0.85 \pm 0.08$ |

## C.3   Details of Real-world Robots

In this section, we detail the real-world robot applications of our method. We deploy synthesized policies on the Unitree Go2 quadruped, designing diverse real-world testing environments to evaluate two key aspects of agent performance: (1) stability during high-speed turning and backward locomotion, and (2) robustness of movements on uneven terrain (mats). Our deployment process consists of four key steps:

- Obtain actor network parameters and corresponding test trajectories from IsaacGym simulations, where the actors are trained using walk-these-ways [14].

- Train **Make-An-Agent** using the acquired training data.

- Generate actor networks from randomly sampled IsaacGym trajectories, covering a variety of training periods.
- Equip the Unitree Go2 quadruped with the generated actors and a pretrained adaptor, enabling it to complete designed, challenging locomotion tasks.

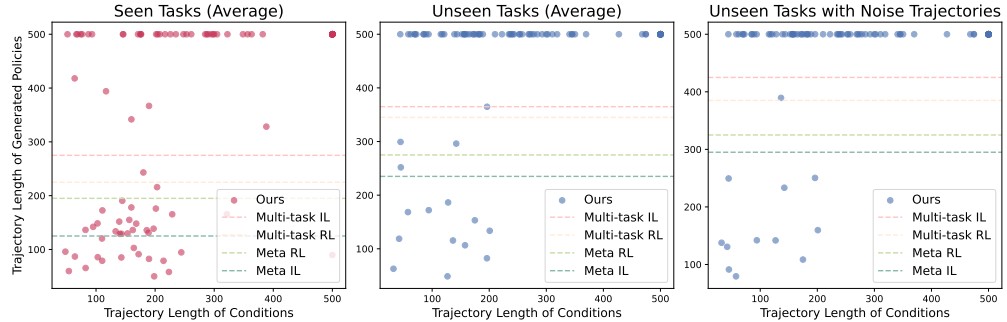

Figure 13: **Correlation between condition trajectories and generated policies.** Trajectory length accurately reflects the effectiveness of the policies compared to the success rate. The maximum episode length in all the tasks is 500 (represents failure).

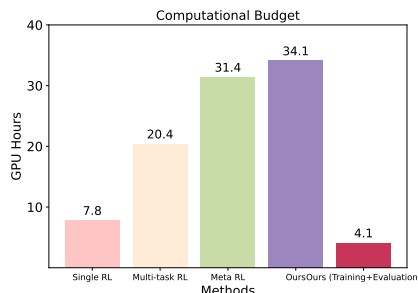

Figure 14: Computational budgets of ours and baselines

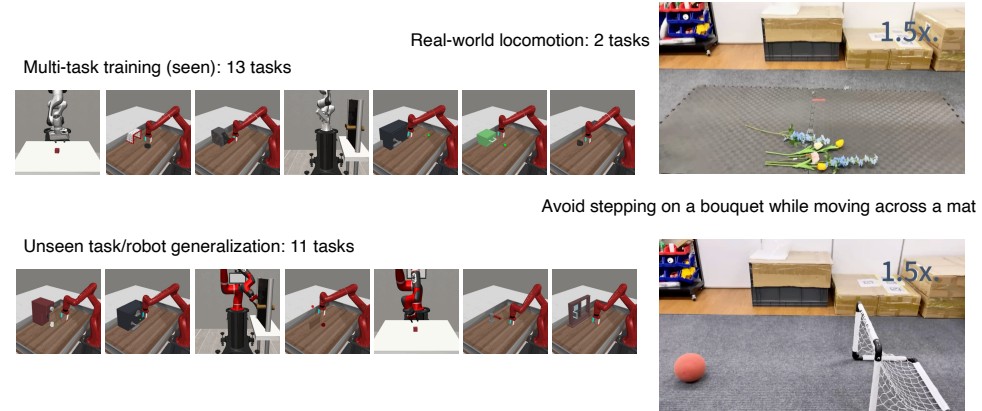

Figure 15: Visualization of experimental tasks

