# OpenReview forum: "Make-An-Agent: A Generalizable Policy Network Generator with Behavior-Prompted Diffusion"
_NeurIPS.cc/2024/Conference — NeurIPS 2024 poster_

### Official Review · Reviewer_YLrf · 2024-07-07

**Soundness:** 3
**Presentation:** 2
**Contribution:** 2
**Rating:** 4
**Confidence:** 4

**Summary:**

Make-An-Agent is a novel policy network generator that uses conditional diffusion models to create control policies based on a single demonstration of desired behaviors. By encoding behavior trajectories into embeddings, Make-An-Agent generates latent policy parameter representations that are decoded into functional policy networks. This method, trained on a dataset of policy parameters and their corresponding trajectories, excels in generating versatile and scalable policies across various tasks, including unseen ones, using few-shot demonstrations. Its robustness and efficiency are demonstrated in both simulation and real-world environments, highlighting its ability to produce high-performing policies even from noisy inputs. This approach bypasses traditional behavior modeling, instead leveraging the inherent correlations in agent behaviors to optimize policy generation without the need for further fine-tuning.

**Strengths:**

1. The paper introduces a novel method that uses conditional diffusion models for policy generation, a significant shift from traditional policy learning methods.
2. Make-An-Agent demonstrates the ability to generate effective policies for a wide range of tasks by conditioning on behavior embeddings, showcasing scalability across different domains.
3. The diffusion-based generator exhibits strong generalization capabilities, producing proficient policies even for unseen behaviors and unfamiliar tasks.
4. The method can generate diverse and resilient policy parameters, maintaining high performance under environmental variability and with noisy input data.
5. The effectiveness of Make-An-Agent is validated not only in simulations but also in real-world robotics, highlighting its practical applicability and efficiency.

**Weaknesses:**

While the paper demonstrates the method's effectiveness in various tasks, a more extensive evaluation across a broader range of environments and conditions could provide a more comprehensive assessment of its robustness and versatility.

The performance of the generated policies heavily relies on the quality of the input demonstrations, which might not always be optimal or available in all scenarios.

The model might overfit to the specific types of tasks and behaviors seen during training, potentially reducing its effectiveness in truly novel and diverse environments.

**Questions:**

How does Make-An-Agent perform when faced with tasks that are vastly different from those it was trained on? Are there any specific limitations in task diversity that the model struggles with?

 What are the computational requirements for training and deploying Make-An-Agent? How does its efficiency compare to traditional reinforcement learning and other policy generation methods?

 How sensitive is the method to the quality of the behavior demonstrations? What happens if the demonstrations contain suboptimal or noisy behaviors?

 Can the approach be scaled to more complex tasks and larger environments? What modifications, if any, would be needed to handle such scalability?

 What are the contributions of the individual components of the method, such as the autoencoder, contrastive learning for behavior embedding, and the diffusion model? Are there ablation studies that show the impact of each component on the overall performance?

 What challenges were encountered during the deployment of the generated policies onto real-world robots? How were these challenges addressed?

 The paper mentions robustness to noisy trajectories, but to what extent can the model handle extreme levels of noise or inaccuracies in the demonstrations?

**Limitations:**

The training process for the autoencoder and diffusion model is likely to be resource-intensive, potentially limiting accessibility for researchers with limited computational resources.

The effectiveness of the generated policies is heavily reliant on the quality of the input demonstrations. Poor or suboptimal demonstrations could adversely affect the performance of the generated policies.

 The evaluation might be limited to specific tasks and environments. A broader range of experiments would be needed to fully assess the model's robustness and versatility across various domains and conditions.

 While the method shows promise in the tested domains, its scalability to more complex tasks and larger environments is not fully explored. Handling such scalability might require further modifications and optimizations.

 There is a risk that the model might overfit to the types of tasks and behaviors seen during training, which could reduce its effectiveness in truly novel and diverse environments.

 While the method has shown success in real-world robotics, the transition from simulation to real-world applications can introduce unforeseen challenges and complexities that need to be addressed comprehensively.

 The paper may lack thorough ablation studies to understand the contributions of individual components of the method, such as the autoencoder, behavior embeddings, and diffusion model.

Although the method claims robustness to noisy trajectories, the extent to which it can handle extreme levels of noise or inaccuracies in the input demonstrations is not fully detailed.

 The sensitivity of the model to various hyperparameters and the process of tuning these parameters for optimal performance is not extensively covered, which can be crucial for practical implementations.

---

> ### Author Rebuttal · Authors · 2024-08-07
>
> We sincerely appreciate Reviewer YLrf for the insightful feedback and the acknowledgment of our work's novelty and empirical effectiveness. Below, we provide a detailed response to address your concerns:
>
> - **W1 and L3: Limited evaluation**: Our experiments cover **3 domains and 23 tasks**, including key control tasks like tabletop manipulation and locomotion control. As acknowledged by Reviewers 8Q7f, 957J, and U6yw, our experiments are more extensive than all the baselines. We also evaluated our approach in real dynamic tasks, including sharp turns, obstacle avoidance on mats, and rapid backward movements. We present more task visualization in Figure 15 in our supplementary material. If you have any further detailed feedback about our experiments, we would be happy to discuss it further.
>
> - **W2, Q3 and L2: Reliance on input demonstrations**: There may be some misunderstanding here. As stated in **Line 187-188**, we use trajectories from RL replay buffer to generate policies. We do not use optimal trajectories or select specific ones, ensuring availability and diversity of conditions. Experiments with noisy trajectories also show that our method can synthesize effective policies under perturbed conditions.
>
> - **W3 and L5: Overfitting to specific tasks and behaviors**: In **Figure 8**, we compare the behaviors used as generation conditions with the behaviors of generated policies, highlighting the differences between them. **Figure 10** demonstrates that our method explores the parameter space more extensively, not overfitting to specific tasks. Additionally, finetuning the model on significantly different environments (Metaworld to Robosuite) proves our generator's effectiveness across domains.
>
> - **Q1: Performance on vastly different tasks**: We present additional generalization experiments on significantly different tasks like soccer, box close, and sweep into in Metaworld. Our method consistently finds optimal policies, outperforming baselines. As tasks with large differences may have significantly different parameter distributions, this issue exists in both prior policy learning methods and our method. Our work aims to minimize adaptation costs as much as possible.
>
> | Method / Success Rate          | Soccer | Box Close | Sweep Into |
> |--------------------------------|--------|-----------|------------|
> | Meta RL with Hypernetwork      | 42.4%  | 55.2%     | 64.3%      |
> | Meta Decision Transformer      | 55.4%  | 52.1%     | 61.5%      |
> | **Ours** (Top 5)               | 69.2%  | 75.3%     | 87.5%      |
> | **Ours** (Best)                | 92.7%  | 87.4%     | 93.8%      |
>
> - **Q2 and L1: Computational requirements**: In **Line 186**, we specify the GPU hours required for collecting policy data for training. In **Lines 425, 434, and 438**, we mention the GPU hours required for training each model. We compare the computational budget in Figure 14 and the below table. Our method requires relatively few GPU hours compared to baselines (much fewer than single RL training). Our training dataset will be released to the community for further valuable research, which we believe contributes to researchers who may have limited resources.
>
> | Method                        | Total Computational Budget (GPU Hours) |
> |-------------------------------|----------------------------------------|
> | Single-task RL                | 7.8                                    |
> | Multi-task Learning                 | 20.4                                  |
> | Meta Learning   | 31.4                                   |
> | **Ours (Including data collection)** | 34.1                          |
> | **Ours (Training/inference/evaluation)** | 4.1                         |
>
> - **Q4 and L4: Scalability**: Our method handles policy networks with parameters ranging from 2e4 to 1.6e5, suitable for common continuous control scenarios. This is both more diverse and larger than the policy networks used in other policy generation methods. For more complex tasks, we can integrate pre-trained visual foundation models as encoders to process image features while using our method to generate actor networks.
>
> - **Q5, L7 and L9: Component contributions and ablation studies**: **Figure 7** shows results without behavior embedding. Since we use a latent diffusion model as the backbone, we can't remove the autoencoder to directly generate the policy parameters, as this would consume a lot of computational cost.
> The three parts of our method are not independent and separable but work together through the autoencoder and behavior embedding to reduce the difficulty of generating in the parameter space and to improve efficiency and effectiveness. Our ablation experiments include detailed studies on behavior embedding and parameter representations. The hyperparameters for our model are based on stable diffusion, with minimal impact on results. All models were trained using a single set of hyperparameters without tuning.
>
> - **Q6 and L6: Real-world deployment challenges**: Deploying policies trained in simulators to real-world environments posed challenges due to the changing terrain and stability. We addressed this by adjusting environment randomness during policy dataset collection, making generated policies more adaptable to real-world dynamics.
>
> - **Q7 and L8: Robustness to noisy trajectories**: As mentioned in **Line 237**, we mention that we use the common setting in adversarial RL by adding Gaussian noise with a standard deviation of 0.1 to actions in test trajectories, which can demonstrate the effect of perturbed trajectories on the generated results within a reasonable range.
>
> In summary, we are grateful to Reviewer YLrf for the detailed feedback. Besides the questions that can be answered within our paper, we will include new discussions from the rebuttal in the appendix to enrich our paper. We hope our response sufficiently addresses your concerns. We eagerly anticipate further discussions.

---

> ### Comment · Area_Chair_kGqK · 2024-08-13
> **Required Action: Please Respond to the Author Rebuttal**
>
> Dear Reviewer YLrf,
>
>
> As the Area Chair for NeurIPS 2024, I am writing to kindly request your attention to the authors' rebuttal for the paper you reviewed.
>
> The authors have provided additional information and clarifications in response to the concerns raised in your initial review. Your insights and expertise are invaluable to our decision-making process, and we would greatly appreciate your assessment of whether the authors' rebuttal adequately addresses your questions or concerns.
>
> Please review the rebuttal and provide feedback. Your continued engagement ensures a fair and thorough review process.
>
> Thank you for your time and dedication to NeurIPS 2024.
>
>
> Best regards,
>
> Area Chair, NeurIPS 2024

---

> ### Author Response · Authors · 2024-08-13
> **Eagerly awaiting your valuable feedback Reviewer YLrf (for the final 12 hours)**
>
> Dear Reviewer YLrf,
>
> As the discussion period draws to a close in 12 hours, we are delighted to have received positive feedback from the other four reviewers and are very eager to ensure that our response has adequately addressed your concerns as well.
>
> We believe that the clarification in our paper would solve your concerns on input demonstrations/overfitting/computational cost/ablation/robustness, along with the additional results and discussions in rebuttal could solve your questions regarding experiments/scalability/real-world deployment.
>
> We deeply appreciate your contribution to the community through your review. Your insights are valuable to us. We eagerly await your response.
>
> Warm Regards,
>
> Paper 8467 Authors

---

### Official Review · Reviewer_8Q7F · 2024-07-11

**Soundness:** 3
**Presentation:** 3
**Contribution:** 2
**Rating:** 5
**Confidence:** 4

**Summary:**

This paper proposes a novel approach to generate policy parameters based on the behavior through diffusion models. The paper leverages the autoencoder to map the parameters of the policy to latent representations. The model demonstrates remarkable generalization abilities on unseen tasks with few-shot demonstrations.

**Strengths:**

* The method is novel. The paper learns behavior embeddings from agent behaviors to capture environmental dynamics and information about tasks. And it leverages such behavior embeddings as conditions for the parameter generation.
* The empirical results are rich and strong. The paper evaluates the model across three environments (two simulations and one real world) and shows strong generalization performance on unseen tasks and for unseen robots.
* This paper also presents well including figures and written structures.

**Weaknesses:**

* The generated policies seem unstable. The generation by diffusion models are diverse but not stable. And the paper doesn't make sure the generated policies are good enough. And the experiments compare the best or top 5 generated policies. How to evaluate which policy is best? Does it require testing all policies and finding which one is best? In C.2, the paper shows the qualification rate of generated policies. The qualification rates are very weak (mostly less than 0.1) on unseen tasks except "drawer close". It's not practical. What if using expert demonstrations?
* Some settings of experiments seem unreasonable. Why fix the initial locations during the training stage and finetuning/adaptation? This might make the baselines or generated policies only work on fixed initial locations. Similarly, why choose highly sub-optimal trajectories? If so, it may use offline RL approaches to finetune the baselines.
* Lack of details of baselines. How to finetune the multi-task RL with fixed trajectories (10/50/100)?

**Questions:**

* See weakness.
* What are the states or what would happen after the first success time? Does the env stop if the task is succeeded?

**Limitations:**

*  The paper has provided several limitations. The paper only generates parameters for simple linear layers. And the learned behavior embeddings are also important.

---

> ### Author Rebuttal · Authors · 2024-08-07
>
> We are grateful to Reviewer 8Q7F's acknowledgment of the novelty, empirical results, and presentation of our paper. Your feedback is very helpful in improving the quality of our work. Below are our detailed responses to each of your questions:
>
> > The generated policies seem unstable. The generation by diffusion models are diverse but not stable.
>
> The instability in the generated policies primarily arises from two factors:
>
> 1. We use trajectories from the replay buffer for generation, which introduces instability in the results. Figure 13 in our supplementary material illustrates the correlation between the effectiveness of condition trajectories and the performance of generated policies. That is reasonable because failed trajectories may not provide sufficient information as prompts for generation.
>
> 2. For unseen tasks, the unfamiliarity and sensitivity in exploring the parameter space lead to greater variability in results. However, compared to other generalization methods that struggle to achieve optimal policies on unseen tasks, our approach is capable of discovering optimal policies in such scenarios.
>
> > How to evaluate which policy is best? Does it require testing all policies and finding which one is best?
>
> To efficiently evaluate policies, we initially validate all generated policies with a single episode under 4 random seeds (**only need 2 minutes**). Subsequently, the qualified policies, sorted by trajectory length from shortest to longest (indicating the used steps for task completion), are tested over 10 episodes to report the final results following all the baselines. Although we need to evaluate all the policies, the GPU hours consumed are significantly less than finetuning required by baselines.
>
> | Task/Average (100 Trajectories)                                             | GPU Hours (100 Trajectories) |
> |---------------------------------------------------|------------------------------|
> | Finetuning (Baselines/Average)                   | 0.40 +- 0.12                          |
> | Inference + Evaluation (100 Policies)| 0.11 +- 0.0                         |
>
> - The qualification rates are very weak (mostly less than 0.1) on unseen tasks except "drawer close". It's not practical. What if using expert demonstrations?
>
> Figure 13 illustrates that the performance of generated policies on unseen tasks is strongly influenced by unfamiliar suboptimal condition trajectories.  Using expert demonstrations can largely improve the qualification rate of generated policies on these "very weak" unseen tasks.
>
> |         Qualification rate  (%)                           |door lock| button press wall |handle press side| faucet open |reach wall |  coffee button
> |---------------------------------------------------|----------|--------------|---------------|---------------|-------------|-------------|
> |**Ours**|0.43|0.35|0.60|0.39| 0.58 | 0.62|
>
> We further compared our method with the best baseline (Meta DT) fine-tuned using expert demonstrations. Our approach still outperforms baselines. We will include the experimental results using expert demonstrations in our paper.
>
> |         Success rate  (%)                           | Unseen 8 tasks (Average)|
> |---------------------------------------------------|----------|
> |Baseline |0.79 +- 0.06|
> |**Ours (Top 5)**|1.0 +- 0.0|
>
> - Why fix the initial locations during the training stage and finetuning/adaptation?
>
> During training, we fix the initialization because we collect multiple policies from a single RL training, which can effectively reduce the computational cost of data collection (instead of changing the initialization each time to collect only one policy). The test trajectories used for finetuning/generating are derived from a single training buffer, thus having the same initialization. For our evaluation results, we conduct experiments in 4 randomly initialized environments over 10 episodes to obtain results. The results indicate that our generated policies do not only work on fixed initial locations.
>
> - Similarly, why choose highly sub-optimal trajectories? If so, it may use offline RL approaches to finetune the baselines.
>
> As mentioned in **Lines 187-188**, we select test data from the SAC training replay buffer within the first 0.5 million timesteps considering that expert demonstrations are not always available and using similar expert trajectories might not provide sufficient diversity.
> We compare our approach with multi-task offline RL methods: skills regularized task decomposition (SRTD). We train the policy using offline training data and then update the policy using test trajectories on unseen tasks. Offline multi-task RL performs worse than imitation learning methods and our approach demonstrates a significant advantage.
>
> |         Success rate  (%)                           | Unseen 8 tasks (Average)|
> |---------------------------------------------------|----------|
> | SRTD |0.61 +- 0.02|
> |**Ours (Top 5)**|0.86 +- 0.07|
>
> [1]Yoo, Minjong, Sangwoo Cho, and Honguk Woo. "Skills regularized task decomposition for multi-task offline reinforcement learning." Advances in Neural Information Processing Systems 2022.
>
> - How to finetune the multi-task RL with fixed trajectories (10/50/100)?
>
> Previous multi-task RL methods have poor performance on unseen tasks. We update the mixture of encoders and critic networks to improve its generalizability. We will include offline multi-task RL as baselines in our paper.
>
> - What are the states or what would happen after the first success time?
>
> In all simulator settings, the environment resets after reaching the maximum episode length. The agent's actions continue selected by the policy network after success. We maintain this environment setting consistent with all baselines.
>
> We thank Reviewer 8Q7F for the detailed and valuable feedback. All the discussions and additional results will be included in the final version. We look forward to further discussions to ensure that our answers address your concerns.

---

> > ### Comment · Reviewer_8Q7F · 2024-08-10
> >
> > Thanks for the response.
> > * collecting data from **fix locations** and **highly sub-optimal** would hugely hurt the baselines' performance. I understand the work has achieved better results with such settings. I'm curious about which factor results in the weak performance of baselines.
> > * In some settings, it's expensive and unsafe to evaluate the policy (like real-world) while it's simpler to collect expert demonstrations with teleoperation systems.
> > * About real-world experiments, the paper seems only to provide some visualizations. What about the success rate and other baselines.

---

> ### Author Response · Authors · 2024-08-10
>
> Dear Reviewer 8Q7F,
>
> Thank you for your time and effort in reviewing our work and providing insightful feedback.
>
> > Collecting data from fix locations and highly sub-optimal would hugely hurt the baselines' performance.
>
> In our rebuttal results, we have demonstrated that our method not only excels in suboptimal data but also significantly outperforms all baselines when using expert demonstrations with random initialization. It's important to note that our method is not an imitation learning approach. Offline replay datasets are widely used in the evaluation of offline RL and meta RL, which is why we selected this evaluation metric.
>
> We fully agree with your suggestion and will include experiments using expert demonstrations in the appendix to fairly compare our method with meta IL baselines.
>
> > It's expensive and unsafe to evaluate the policy (like real-world) while it's simpler to collect expert demonstrations with teleoperation systems.
>
> In real-robot experiments as mentioned in our paper, we utilize the IsaacGym simulator to collect policies and trajectories and also evaluate generated policies before deploying them on the real robot, which ensures that we conduct evaluations at a relatively low cost.
>
> There is no contradiction between using simulation for initial evaluation and using expert demonstrations for generation. If optimal real-world data is available, using it to generate policies would certainly be preferable. Our experiments aim to demonstrate that policies generated and evaluated in simulation can still yield excellent results in real-world scenarios.
>
> > About real-world experiments, the paper seems only to provide some visualizations. What about the success rate and other baselines.
>
> Thank you for your suggestions. Real-world visualization is the most straightforward way to observe the performance of policies, especially since designing success rate metrics for real-robot locomotion tasks is challenging as shown in prior works. Additionally, none of the baselines we listed include real-world experiments, and the network sizes used in those baselines are not applicable to real-robot scenarios.
>
> We evaluate our generated policies against offline RL methods in low-speed testing and high-speed testing following [1]. The results demonstrate that the policies generated by our method exhibit better stability and performance on real robots.
>
> | Low-Speed Testing | IQL | BC | Ours |
> |---|---|---|---|
> |Reward |  11.9 |  14.4 | 24.8 |
>
> | High-Speed Testing | IQL | BC | Ours |
> |---|---|---|---|
> |Reward |  10.4 |  10.7 | 20.9 |
>
> [1] Margolis, Gabriel B., and Pulkit Agrawal. "Walk these ways: Tuning robot control for generalization with multiplicity of behavior." Conference on Robot Learning. PMLR, 2023.
>
> We hope our responses address your concerns, and we are open to further discussion. Thank you once again for your valuable feedback.
>
> Warm Regards,
>
> Paper 8467 Authors

---

> > ### Comment · Reviewer_8Q7F · 2024-08-13
> >
> > Thanks for the detailed response. I maintain my positive rating.

---

> ### Author Response · Authors · 2024-08-13
> **Thank Reviewer 8Q7F for your inspiring reply!**
>
> Dear Reviewer 8Q7F,
>
> Thank you for your reply and understanding. We will update our paper accordingly based on your and other reviewers' comments.
>
> We sincerely appreciate your positive comments and the discussions during rebuttal, which are of great help of improving our paper and also contribute to the community.
>
> Best wishes!
>
> Paper 8467 Authors

---

### Official Review · Reviewer_Pxd2 · 2024-07-13

**Soundness:** 4
**Presentation:** 4
**Contribution:** 4
**Rating:** 7
**Confidence:** 4

**Summary:**

This paper proposes the Make-An-Agent architecture, which synthesizes a policy neural network from an input trajectory. Make-An-Agent utilizes a parameter and behavior embedding. The behavior embedding is trained with the mutual information between the trajectory and the successful part of the trajectory. The behavior embedding is then used in a diffusion model to generate the policy parameters. This methodology is validated in MetaWorld, Robosuite, and quadrupedal locomotion environments. In MetaWorld and RoboSuite, Make-An-Agent outperforms baselines RL and IL baselines. Further experiments analyze how the trajectories and policies produced by Make-An-Agent differ from those directly trained through RL.

**Strengths:**

1. The paper represents a novel take on learning multi-task policies by directly generating the parameters of a policy network from a demonstration. This is different from prior Meta-RL approaches that adapt existing networks.

1. The paper has extensive empirical comparisons to a variety of baselines from RL and IL. The proposed method outperforms baselines in MetaWorld and Robosuite by a large margin.

1. Make-An-Agent is capable of generating policies that are more diverse (Figure 8) and more robust (Figure 6) than baselines.

1. The behavior embedding is an important aspect of the Make-An-Agent architecture as demonstrated by Figure 7.

1. The paper analyzes the impact of important settings on the system's performance with the demonstration length, policy size and number of policy parameter sets used for training.

1. Implementation details are thoroughly described throughout the paper and in supplementary sections B and C.

**Weaknesses:**

1. Table 1 only reports the best and top 5 generated policies. However, all policies had to be evaluated in the environment to decide what these top policies are. This is an unrealistic assumption that weakens the significance of the result. The paper must analyze in more detail the distribution of performance between generated policies and compare the full statistics of all generated policies to baselines for a fair comparison.

1. A weakness of directly generating policy parameters is the challenge of scaling up to larger policy networks. For example, policies operating from egocentric visual inputs in 3D spaces typically have tens of millions of parameters. The core idea of directly generating policy parameters is likely incapable of scaling to this setting of 1000x more policy parameters.

**Questions:**

1. What fraction of generated policies outperform the baselines in Table 1? And what is the average policy performance between all generated policies, not just the top 5? See my point (1) under weaknesses above.

1. Given that the results in Figure 11b already show performance suffering with increased policy size, how can Make-An-Actor scale to more complex tasks that require higher parameter count policies?

**Limitations:**

Yes, limitations are discussed in Section 6.

---

> ### Author Rebuttal · Authors · 2024-08-07
>
> We sincerely appreciate the positive feedback from Reviewer Pxd2 on the originality, overall quality, and significance of our work. The valuable comments and suggestions from Reviewer Pxd2 are of great help to improve the quality of our work. Detailed responses regarding each problem are listed below.
>
> > All policies had to be evaluated in the environment to decide what these top policies are. This is an unrealistic assumption that weakens the significance of the result. The paper must analyze in more detail the distribution of performance between generated policies and compare the full statistics of all generated policies to baselines for a fair comparison.
>
> The evaluation metrics we reported are chosen primarily for the following reasons:
> 1. **Feasibility and efficiency of evaluation:** Our method can generate one policy per trajectory, whereas previous methods can only train one policy through fine-tuning with multiple trajectories. As mentioned in our paper, the key difference is that we do not need to use trajectories for finetuning. Instead, we generate policies and evaluate them with one episode to determine their effectiveness. The final results are then reported after evaluation over 10 episodes, consistent with the baselines. The GPU hours consumed by the inference process and evaluation for 100 policies are listed below. We also utilize IsaacGym simulator to evaluate policies before deploying them on the real robot, ensuring evaluation at a relatively low cost.
>
>
> | Task/Average (100 Trajectories)                                             | GPU Hours (100 Trajectories) |
> |---------------------------------------------------|------------------------------|
> | Finetuning (Baselines/Average)                   | 0.40 +- 0.12                          |
> | Inference + Evaluation (100 Policies)| 0.11 +- 0.0                         |
>
> 2. **Effectiveness of all generated policies:** We add a set of results in Figure 13 in our supplementary material to present the performance of all the generated policies compared with baselines. Since we do not actually need to use all the generated policies (just as we do not need to use all generated results), but only need to obtain one optimal policy to consistently complete the task, our method provides a new pathway for addressing unseen tasks. This is in contrast to baselines, which cannot achieve good unseen generalization even with finetuning. We also believe that using the generated multiple optimal policies to learn a mixture of experts for decision-making is a potential future direction.
>
>
> > A weakness of directly generating policy parameters is the challenge of scaling up to larger policy networks. For example, policies operating from egocentric visual inputs in 3D spaces typically have tens of millions of parameters. The core idea of directly generating policy parameters is likely incapable of scaling to this setting of 1000x more policy parameters.
>
> This is indeed a very worthwhile discussion. Using visual encoders to handle image or 3D point cloud inputs is already a common approach. We can apply our method to generate actor or critic networks but with inputs processed through existing pre-trained visual encoders to manage different environmental inputs. Our paper covers a broad range of policy network sizes, from 2e4 to 16e4 parameters, which is among the widest ranges addressed by existing network generation methods. Additionally, we propose other potential solutions for addressing complex networks in our responses below.
>
> > Given that the results in Figure 11b already show performance suffering with increased policy size, how can Make-An-Actor scale to more complex tasks that require higher parameter count policies?
>
> In Figure 11b, the decline in performance with larger policy sizes is likely due to the autoencoder's inability to provide effective parameter representations for large networks, which significantly impacts the effectiveness of the generated policies (for fairness, we did not modify any model parameters for large networks). When we scale up the hidden size of the autoencoder from 1024 to 2048, policies with 256 hidden sizes achieve very similar results to those with hidden sizes of 128. Therefore, an effective strategy for handling larger networks might be to use multiple autoencoders to encode different layers of the network separately. Alternatively, generating different layers or portions of larger policy networks tailored to specific task features could facilitate faster adaptation to various complex downstream tasks. We believe that exploring how to generate larger networks is a highly promising potential direction.
>
> Thank you again for carefully reviewing our paper and providing very constructive suggestions. We will incorporate the above discussions into our final version. We hope that this addresses your concerns, and we are happy to engage in further discussion.

---

> ### Comment · Reviewer_Pxd2 · 2024-08-11
>
> Thank you for the response. While I still appreciate the novelty of the approach, like other reviewers, I am also concerned about the evaluation criteria for selecting the best policies. The GPU hours comparison in the rebuttal is not a representative comparison because for Make-An-Agent, this also includes evaluation costs, which could be prohibitively expensive depending on the environment, as in the real world or a slower simulator. Figure 13 in the rebuttal is also unclear. Which environment are these results for? Why compare episode length in the tasks? A clearer comparison would be to generate a histogram of success rates, not trajectory lengths, for the Make-An-Agent generated policies across all condition trajectories.

---

> ### Author Response · Authors · 2024-08-11
> **Thank Reviewer Pxd2 for your supportive feedback!**
>
> Dear Reviewer Pxd2,
>
> Thank you for your prompt reply and constructive suggestions.
>
> * In our GPU hours comparison, we have included the evaluation and inference costs. For very slower simulators, we strongly agree that filtering policies by selecting condition trajectories that are close to optimal may reduce evaluation time costs. Even if more interaction time is required, the evaluation cost in most environments is still significantly lower than the cost of fine-tuning.
>
> * In our real-robot experiments, as discussed in our paper, we utilize the IsaacGym simulator to collect policies and trajectories and also evaluate the generated policies before deploying them on the real robot, which ensures that we conduct evaluations at a relatively low cost. Our experiments aim to demonstrate that policies generated and evaluated in simulation can still achieve excellent results in real-world scenarios.
>
> * Figure 13 presents results from the door unlock (seen) and coffee button (unseen) environments, both of which exhibit medium performance across all seen/unseen tasks.
>
> * We chose to use trajectory length as it more accurately reflects the performance of the condition trajectories compared to the success condition. For example, two trajectories might both achieve success, but their lengths could differ, with shorter trajectories indicating closer to optimal and more efficient performance. Using trajectory length to evaluate policies also makes it easier to compare with condition trajectories.
>
> * We fully acknowledge your suggestion, and we provide a table showing the generated policies' success rates across all condition trajectories for all seen/unseen tasks. We can see that on seen tasks, our method can generate more than 55% of policies that perform better than the best baselines. On unseen tasks, on average at least 35% of the policies perform better than the best baselines. This fully demonstrates the superiority of our method.
>
> * Additionally, if we select trajectories with success signals for policy generation, in seen tasks, more than 82% of the policies outperform the baselines, and in unseen tasks, more than 50% do so. For environments where evaluation is challenging, using success trajectories can also more efficiently yield optimal policies. We will include these results as histograms in the final version of our paper.
>
> | Average success rate (seen) | Generated policies across all trajectories |
>  | ------ | ----- |
> |   0%       |    21.2%     |
> |   0-60%       |   5.2%      |
> |   60-80%       |  17.7%       |
> |   80-100%       |    26.5%     |
> |   100%       |    29.4%     |
>
> | Average success rate (unseen) | Generated policies across all trajectories |
>  | ------ | ----- |
> |   0%       |    27.8%     |
> |   0-60%       |   12.0%      |
> |   60-80%       |   28.2%      |
> |   80-100%       |    19.6%     |
> |   100%       |    12.4%     |
>
> | Average success rate (seen) | Generated policies across success trajectories |
>  | ------ | ----- |
> |   0%       |    3.2%     |
> |   0-60%       |   6.9%      |
> |   60-80%       |  11.2%       |
> |   80-100%       |    8.4%     |
> |   100%       |    70.3%     |
>
> | Average success rate (unseen) | Generated policies across success trajectories |
>  | ------ | ----- |
> |   0%       |    10.7%     |
> |   0-60%       |   17.8%      |
> |   60-80%       |   21.4%      |
> |   80-100%       |    18.9%     |
> |   100%       |    31.2%     |
>
> **In summary**, we agree that the evaluation cost could become a weakness of our method when generating multiple policies. However, it is undeniable that the broader exploration of parameter spaces offers more effective solutions for policy generalization compared to prior works. We also believe that using these multiple policies for a mixture of experts could be a promising direction.
>
> Thank you for acknowledging the novelty of our work and for providing such insightful feedback. We welcome any further discussions and greatly value the opportunity to continue improving our research.
>
> Best Regards,
>
> Paper 8467 Authors

---

### Official Review · Reviewer_957J · 2024-07-13

**Soundness:** 3
**Presentation:** 4
**Contribution:** 3
**Rating:** 6
**Confidence:** 4

**Summary:**

In this work, the authors present Make-An-Agent, which is a method to generate policy parameters given a few intended trajectories from a task. The proposed method is straightforward, which makes it better that it seems to work in the experiments in the work. The method first generates a large dataset of policies as well as their rollout traces, all parametrized in the same way, using standard RL algorithms. Then, the authors learn an autoencoder on top of the policy parameters. Finally, the method learns a trajectory conditional generator using diffusion for creating policy latents. Putting them together, on a new environments the authors generate some trajectories (here, by running SAC for 0.5M steps), and then generate a new policy using the trajectory latent which gets translated to a policy latent by the diffusion model.

The experiments are done on a real robot locomotion task as well as two manipulation task suites in simulation. In all of these cases, the best generated policies perform well in the experiments compared with meta-learning approaches.

**Strengths:**

+ The approach is novel, and the algorithm and the released code are both quite simple.
+ The experiments have a wide breadth, and showing the applicability in multiple different domains show the potential of this method.
+ While the dataset is large (1000+ policies per task) it is not excessively large to be prohibitive.

**Weaknesses:**

- The reported metric (best generated policy performance, avg top 5 generated policy performance) both seem _fudged_. It seems like the authors are generating a large number of policies, and then picking the best generated policy out of them to get their numbers. This seems dishonest, as for a large number N of generated policies, every method will get a very high score.
- While the authors report the "qualification rate" i.e the % of policies getting 100% success rate in test, I am not sure how this works. As they report the "best policy score", and since qualification rate is > 0 for all tasks, why is figure 5 (ours) not all 100%? What am I missing?
- The drop between top-1 and top-5 is concerning, since I cannot tell whether the authors got lucky finding a perfect policy from the long tail.
- The authors don't mention the overall compute overhead and how it compares with the baselines, which I believe is a strong consideration for such a method.

**Questions:**

- Please detail the compute budget for each of the baselines as well as your method.
- If possible, please release the dataset as well as the pretrained networks with your code.

**Limitations:**

- Generating larger networks will be more difficult, so I wish the authors took advantage of any special properties of network weights while training.
- The comparison metric is fuzzy and can seem unfair.

---

> ### Author Rebuttal · Authors · 2024-08-07
>
> We sincerely appreciate Reviewer 957J for your acknowledgment of our idea's novelty and method's applicability. Thank you for your valuable comments and suggestions, which are of great help to improve the quality of our work. We carefully answer each of your concerns as below.
> > The reported metric (best generated policy performance, avg top 5 generated policy performance) both seem fudged. It seems like the authors are generating a large number of policies, and then picking the best generated policy out of them to get their numbers.
>
> Regarding the reported metrics, we primarily consider the following perspectives :
> 1. **Generate vs. Finetune**: Our method can generate one policy per trajectory, while previous methods can only train one policy through fine-tuning with multiple trajectories. Even with multiple finetuning attempts to get the best result, no method can achieve a very high score. (The baseline results we report are the best models obtained during finetuning)
>
> 2. **Fairness and Efficiency**: The test trajectories we use are exactly the same as those used for the baselines. The difference is that we do not need to use trajectories for finetuning but generate policies and evaluate them with one episode to determine if the generated policies are effective. The GPU hours consumed by the inference process and evaluation for 100 policies are listed below.
>
> | Task/Average (100 Trajectories)                                             | GPU Hours (100 Trajectories) |
> |---------------------------------------------------|------------------------------|
> | Finetuning (Baselines/Average)                   | 0.40 +- 0.12                          |
> | Inference + Evaluation (100 Policies)| 0.11 +- 0.0                         |
>
> 3. **Usage**: Our goal is to ultimately generate an optimal policy with few trajectories, similar to all previous policy learning methods, but our method achieves better performance. Although we obtained many policies, this paper does not further discuss what can be done with the large number of generated policies. They could potentially be used for learning the mixture of experts, which we believe is a promising future direction.
>
> > While the authors report the "qualification rate", i.e, the % of policies getting 100% success rate in test, I am not sure how this works. As they report the "best policy score", and since qualification rate is > 0 for all tasks, why is figure 5 (ours) not all 100%? What am I missing?
>
> To efficiently evaluate policies and save on evaluation costs, we initially validate all generated policies with a single episode under 4 random seeds (consuming very little GPU time). Subsequently, the qualified policies, sorted by episode length from shortest to longest (indicating the time spent on task completion), are tested over 10 episodes to report the final results. Therefore, some policies may occasionally fail in certain episodes, resulting in a success rate that does not reach 100%.
> It should be noted that evaluating 10 episodes follows the standard metrics of all common baselines, ensuring fairness with other papers. We will add these detailed explanations of the evaluation process in the appendix.
>
> > The drop between top-1 and top-5 is concerning, since I cannot tell whether the authors got lucky finding a perfect policy from the long tail.
>
> Based on the evaluation details provided above, this explains why the performance of the top 5 policies may differ from that of the best policy. The performance can fluctuate across multiple episodes due to the random initialization of the environment, which is a normal phenomenon also observed in single RL policy learning.
>
> > The authors don't mention the overall compute overhead and how it compares with the baselines, which I believe is a strong consideration for such a method. Please detail the compute budget for each of the baselines as well as your method. If possible, please release the dataset as well as the pretrained networks with your code.
>
> In **Line 186**, we state the GPU hours for collecting policy data for training. In **Line 425, 434, and 438**, we mention the GPU hours for training each model. **Our training time requires relatively few GPU hours compared to other methods (much fewer than single RL training)**. All of our datasets and pretrained models will be released to the community for further valuable research, which we believe contributes to researchers who may have limited resources.
>
> | Method                        | Total Computational Budget (GPU Hours) |
> |-------------------------------|----------------------------------------|
> | Single-task RL                | 7.8                                    |
> | Multi-task Learning                 | 20.4                                  |
> | Meta Learning   | 31.4                                   |
> | **Ours (Including data collection)** | 34.1                          |
> | **Ours (Training/inference/evaluation)** | 4.1                         |
>
> > Generating larger networks will be more difficult, so I wish the authors took advantage of any special properties of network weights while training.
>
> This is a very insightful topic for discussion. In our method, we choose to encode the parameters of each layer of the network, leveraging the properties of the policy network. To handle larger networks, we believe several approaches can be considered: generating parameters for each layer individually, generating parameters for only parts of networks, or utilizing pretrained task representations to reduce the dimension of inputs to the network, allowing smaller actor networks to tackle complex tasks. These are all promising future directions worth exploring, and we will include this discussion in the appendix.
>
> Thanks again for reviewing our paper carefully and providing very constructive suggestions. We hope the above resolves your concerns, and we are glad to have any further discussions.

---

> > ### Comment · Reviewer_957J · 2024-08-13
> >
> > Thank you for the response and the discussion. After reading this rebuttal, the only point of contention that remains for me is how efficiency is reported. In the real world (which is where I study robotics) the majority of time and energy is spent on evaluating the robot in a real environment, which is a lot more expensive than simple GPU hours. Since the number of runs required is not disambiguated from GPU hours needed to train/generate/fine-tune, I am keeping my current score.
> >
> > I still recommend acceptance of the paper, which would be a strong accept with:
> > 1. Release of full code and pretrained model library, and
> > 2. Disambiguating between GPU hours needed for training (done offline) and rollouts/evaluation needed for model selection (done online.)

---

> ### Comment · Area_Chair_kGqK · 2024-08-13
> **Required Action: Please Respond to the Author Rebuttal**
>
> Dear Reviewer 957J,
>
>
> As the Area Chair for NeurIPS 2024, I am writing to kindly request your attention to the authors' rebuttal for the paper you reviewed.
>
> The authors have provided additional information and clarifications in response to the concerns raised in your initial review. Your insights and expertise are invaluable to our decision-making process, and we would greatly appreciate your assessment of whether the authors' rebuttal adequately addresses your questions or concerns.
>
> Please review the rebuttal and provide feedback. Your continued engagement ensures a fair and thorough review process.
>
> Thank you for your time and dedication to NeurIPS 2024.
>
>
> Best regards,
>
> Area Chair, NeurIPS 2024

---

> ### Author Response · Authors · 2024-08-13
> **Thank Reviewer 957J for your replay and suggestions!**
>
> Dear Reviewer 957J,
>
> Thank you for your constructive suggestions and understanding. We would like to provide further clarification on the concerns you raised:
>
> > In the real world, the majority of time and energy is spent on evaluating the robot in a real environment, which is a lot more expensive than simple GPU hours.
>
> We completely agree with this point. To ensure a relatively low cost on evaluations, in our real-robot experiments, we utilize the IsaacGym simulator to collect policies and trajectories and evaluate the generated policies before deploying them on the real robot. Our experiments aim to demonstrate that policies generated and evaluated in simulation can still achieve excellent results in real-world scenarios.
>
> For tasks where it is not feasible to evaluate in simulations, we try to discuss alternative methods inspired by other reviewers for utilizing multiple policies in decision-making, such as majority voting and using the mixture of experts, to improve the robustness and stability of real-robot decisions. We believe these are very promising directions and hope our work will inspire the community to explore more.
>
> > Release of full code and pretrained model library
>
> As we have promised, we will release all datasets, pretrained models, and training code in our final version. We greatly appreciate your support.
>
> > Disambiguating between GPU hours needed for training (done offline) and rollouts/evaluation needed for model selection (done online).
>
> Based on your suggestion, we will provide a detailed report on the computational costs associated with each part of our method. Additionally, we will include a separate discussion of the evaluation costs and alternative approaches in the limitations section.
>
> Thank you again for your positive comments and valuable suggestions regarding real-world evaluation, which have greatly helped us further discuss the weaknesses and improve the quality of our paper. We appreciate the time and effort you’ve put into reviewing and the rebuttal process. We will supplement our paper based on all of these discussions.
>
> Best wishes,
>
> Paper 8467 Authors

---

### Official Review · Reviewer_U6yw · 2024-07-15

**Soundness:** 3
**Presentation:** 3
**Contribution:** 3
**Rating:** 6
**Confidence:** 3

**Summary:**

This paper presents Make-An-Agent, a conditional diffusion model that generates policy parameters based on demonstration of target behaviors. The authors propose an autoencoder to encode policy network parameters into compact latent representations. The behavior embeddings are learned using contrastive learning between long-term trajectories and their success or future states. Then, a conditional diffusion model is used to generate policy latent code conditioned on the learned behavior embedding, which can then be decoded to policy parameters via the pretrained decoder. Extensive results in simulated environment including MetaWorld and Robosuite demonstrate the effectiveness of the proposed method. Also, there is real-world deployment on quadrupedal locomotion.

**Strengths:**

- The paper is well-written and easy to follow
- The experimental analysis is pretty thorough.

**Weaknesses:**

- Despite the experiments are overall complete, there are many more interesting aspects to look at, which may further strengthen the paper. Please check questions for more details.
- The real-world experiment has very limited results presented.

**Questions:**

- Is there any way to roughly predict the performance of the generateed parameters without actually evaluating them? For example, assuming the learned data distribution nicely reflects the given behavior, the generation distribution in the policy parameter space should somewhat follow the mode that gives good performance; perhaps something similar to the "Parameter distinction" experiments but among the generation from the proposed model only can be interesting.
- Is any of the statistics of the generated policy parameters useful? For example, does the mean still a valid policy parameter? Is the variance of the generated parameters correlated with how "out-of-distribution" the behavior is?
- It would be interesting to briefly look at the geometry in the behavior embedding space. For example, how does small perturbation in the behavior embedding affect the performance of the generated policy parameters. Or even more interesting, is it possible to compose behaviors by messing around the conditioning of the diffusion model, which then gives a compositional policy?
- Could the authors report the performance distribution for all generations (as opposed to the best or top 5)? This can probably provide some insight into the generation behavior and what distribution (w.r.t. the data distribution in generative modeling not performance distribution) is actually learned by the model. Also, it would be interesting to see the performance distribution in seen and unseen tasks.
- For ablation of policy network, could the authors discuss how architectural difference (beyond hidden size; e.g., constructing policy as very small recurrent network or transformer) can potentially affect the results?
- In real-world evaluation, could the authors provide more details about the behaviors and policies in trainnig dataset and testing?
- Given the proposed method being a generator of the policy (which can easily produce many models at a time), it would be also interesting to check the mixture of expert setup. For example, will using something simple like majority voting in generated policies improves robustness in "Adaptability to environmental randomness on seen tasks"?
- Could the authors report some failure cases? Or is there any failure mode like certain type behaviors/demonstrations always lead to bad-performing generated policies?

**Limitations:**

Limitations are discussed in the paper.

---

> ### Author Rebuttal · Authors · 2024-08-07
>
> We thank Reviewer U6yw for the positive comments to our writing and experiment thoroughness. Your questions are instrumental in helping us improve the quality of our paper. We have provided detailed responses point by point below.
>
> > Is there any way to roughly predict the performance of the generateed parameters without actually evaluating them?
>
> **Feasibility and efficiency of evaluation:** We generate policies and evaluate them with only one episode to determine their effectiveness. The final results are then reported after evaluation over 10 episodes, consistent with the baselines. The GPU hours consumed by the inference process and evaluation for 100 policies are listed below. We ensure a very low cost for evaluation.
>
> | Task/Average (100 Trajectories)                                             | GPU Hours (100 Trajectories) |
> |---------------------------------------------------|------------------------------|
> | Finetuning (Baselines/Average)                   | 0.40 +- 0.12                          |
> | Inference + Evaluation (100 Policies)| 0.11 +- 0.0                         |
>
> > Is any of the statistics of the generated policy parameters useful? Is the variance of the generated parameters correlated with how "out-of-distribution" the behavior is? Is there any failure mode like certain type behaviors/demonstrations always lead to bad-performing generated policies?
>
> Figure 13 in our supplementary material illustrates the relationship between condition trajectories and the performance of generated policies. It can be observed that when the condition trajectories are effective, the generated policies tend to perform better. When the behaviors used for synthesized conditions deviate significantly from the correct actions, the generated policies tend to fail. That is reasonable because failed trajectories may not provide sufficient information as prompts for generation. While condition trajectories do show some correlation with generated policies, it is difficult to use them as a metric to directly judge the performance of generated policies.
>
> > How does small perturbation in the behavior embedding affect the performance of the generated policy parameters. Is it possible to compose behaviors by messing around the conditioning of the diffusion model, which then gives a compositional policy?
>
> We also show the impact of noisy trajectories on the generated results for unseen tasks in Figure 13, demonstrating a significant advantage over the baselines. For combined trajectories, our method can not generate compositional policies, due to the absence of multi-task compositional policies in our training dataset. However, this presents an interesting possibility that could offer new insights into exploring the parameter space for multi-task learning.
>
> > Could the authors report the performance distribution for all generations (as opposed to the best or top 5)?
>
> As shown in Figure 13. Note that we choose average trajectory length as a metric because it more accurately reflects the effectiveness of the policies compared to the success rate. For example, policies with the same 80% success rate can exhibit significant differences in task efficiency.
>
> > For ablation of policy network, could the authors discuss how architectural difference (beyond hidden size; e.g., constructing policy as very small recurrent network or transformer) can potentially affect the results?
>
> The choice of policy architectures determines the type of network parameter data collected for the training dataset. In our experiments, we used MLP policy structures with varying layer numbers, resulting in parameter counts ranging from 2e4 to 16e4 across three domains. The architecture variation did not affect the overall effectiveness in different domains. We believe that our method could also be applied to generate RNN or Transformer policies.
>
> > In real-world evaluation, could the authors provide more details about the behaviors and policies in training dataset and testing?
>
> In our locomotion tasks, we collect the training dataset using the IsaacGym simulator. We train policies on flat ground to move forward stable without any obstacles and changes in terrain geometry, while randomizing dynamics parameters such as payload mass, motor strength, and gravity offset, to obtain policies under different environment settings.
> In the real-world deployment, we apply the generated policies to the Unitree Go2 Edu robot, issuing commands to complete behaviors such as sharp turns, obstacle avoidance on mats, and rapid backward movements. We will update the details of these real-world experiments in the appendix.
>
> > Given the proposed method being a generator of the policy (which can easily produce many models at a time), it would be also interesting to check the mixture of expert setups.
>
> It is a valuable idea that can help us better utilize the large number of generated policies. We employed a straightforward approach by dividing the action space into over 20 discrete intervals. Then we performed majority voting to select the most frequently voted interval. The actions output by policies voting for this interval were averaged to determine the final action.
> We conduct experiments on three unseen tasks and evaluate using adversarial RL settings with random noise in the range of 0.1 added to actions on 4 random initializations. The results show that **multiple synthesized expert policies can significantly improve robustness**.
> | Robustness to random noise (sr) | door close | coffee button | reach wall|
> |-----|----|----|----|
> | Single RL | 0.32 | 0.44 | 0.52 |
> | Ours (Generated Policies)  | 0.43 | 0.56 | 0.55 |
> | Ours (mixture of experts) | 0.89 | 0.82 | 0.79 |
>
> We sincerely appreciate your thorough and detailed feedback and hope our replies address all your concerns. All the discussions and experimental results will be included in the final version to further enhance our paper's quality. We look forward to further discussion.

---

> ### Comment · Area_Chair_kGqK · 2024-08-13
> **Required Action: Please Respond to the Author Rebuttal**
>
> Dear Reviewer U6yw,
>
>
> As the Area Chair for NeurIPS 2024, I am writing to kindly request your attention to the authors' rebuttal for the paper you reviewed.
>
> The authors have provided additional information and clarifications in response to the concerns raised in your initial review. Your insights and expertise are invaluable to our decision-making process, and we would greatly appreciate your assessment of whether the authors' rebuttal adequately addresses your questions or concerns.
>
> Please review the rebuttal and provide feedback. Your continued engagement ensures a fair and thorough review process.
>
> Thank you for your time and dedication to NeurIPS 2024.
>
>
> Best regards,
>
> Area Chair, NeurIPS 2024

---

> > ### Comment · Reviewer_U6yw · 2024-08-13
> > **Thanks for the rebuttal**
> >
> > Thanks for the rebuttals. The mixture-of-experts result is really nice. It would be nice to include this result with the computational overhead of doing so (which I assume is pretty marginal) in the final version. For Fig.13 (re "the statistics of the generated policy parameters"), I was previously referring to experimenting in the parameter space but this result also looks nice. Lastly, for architecture, I don't think the use of other architectures like RNN or transformer (on non-reactive or partially-observable tasks) is trivially extendable; but I am not 100% positive.
> > Overall, the new results look nice. I will keep my rating.

---

> ### Author Response · Authors · 2024-08-13
> **Thank you for your inspiring reply!**
>
> Dear Reviewer U6yw,
>
> Thank you for your thoughtful feedback and understanding! We further discuss the points you raised.
>
> > mixture-of-experts computational overhead
>
> When using the mixture-of-experts approach for decision making, while it involves using multiple policies to output actions, the inference cost for these policies is very low. As a result, the computational cost is less than 1/10 of the baseline fine-tuning cost. We will include detailed statistics on this in the final version of our paper.
>
> > the statistics of the generated policy parameters in parameter space
>
> Given the high dimensionality of the parameter space, we primarily compare the trajectories obtained from deployed policies. Directly averaging the generated parameters does not yield ideal results, mainly because the parameter distribution in the latent parameter space and the parameter space are not smooth.
>
> > the use of other architectures like RNN or transformer
>
> If we use RNN as the policy backbone, and the parameter count of the RNN is not larger by an order of magnitude, it is feasible to use our model to generate RNNs. Our method directly flattens the network parameters into a vector before encoding and generating, without imposing special requirements on the network structure during the generation process.
>
> We sincerely appreciate your positive comments and insightful discussions. The constructive suggestions for practical applications are crucial for improving the quality of our paper, and we will ensure that all discussions and new results are included in the final version.
>
> Best wishes!
>
> Paper 8467 Authors

---

### Author Rebuttal · Authors · 2024-08-07

## General Response
### **Summary of Review and Highlights**
We sincerely thank all reviewers for their insightful comments, valuable questions, and helpful suggestions.

We appreciate the positive feedback from all reviewers regarding our paper's **presentation (Reviewers U6yw, 8Q7F)**, **idea novelty (Reviewers 957J, Pxd2, 8Q7F, YLrf)**, **strong empirical performance (Reviewers 957J, Pxd2, 8Q7F, YLrf)**, and **generalization ability of our method (Reviewers Pxd2, 8Q7F, YLrf)**. We are particularly grateful to **Reviewers 8Q7F, 957J, U6yw, and Pxd2 for highlighting the thoroughness and completeness of our experiments**. Our method introduces a novel generative framework for policy learning that not only offers new insights for downstream generalization but also demonstrates impressive effectiveness across various scenarios.

### **Reviewer Concerns**
We carefully consider the reviewers' feedback and responded to each question with detailed explanations and additional experimental results. Here are our responses to the main concerns:

**Concerns about Computational Cost (Reviewers YLrf, 8Q7F, 957J):** As stated in our paper, the computational cost of training our method is even lower than that of single RL training, with the main GPU hours allocated to collecting data for policy networks.  The overall computational time of our method is comparable to baselines as shown in Figure 14. As one of our contributions, we will open-source all models and datasets to support further valuable research by the community.

**Concerns about Scalability (Reviewers YLrf, 957J, Pxd2):** We demonstrate the scalability of our method by showing the parameter number used across different domains, which is more scalable than existing network generation methods. We also discuss various approaches to address more complex networks, which will be included in the appendix.

**Concerns about Evaluation Metrics (Reviewers 8Q7F, 957J, U6yw, Pxd2):** We explain the differences between our approach and prior policy generalization methods (specifically, our approach does not require finetuning) and highlight the efficiency of our evaluation phase. We clarify the choice of evaluation metrics from multiple perspectives and analyze the performance of all generated policies in Figure 13 in our supplementary material.

### **Additional Results**
* We present the relationship between the effectiveness of generated policies and condition trajectories, as shown in Figure 13.
* The computational budgets compared with baselines are presented in Figure 14.
* In response to Reviewer YLrf, we show generalization results on more diverse unseen tasks.
* In response to Reviewer 8Q7F, we include results of policy generation using expert demonstrations and compare them with offline RL baselines.
* In response to Reviewer U6yw, we demonstrate the improvement in robustness of our generated policies using a majority vote approach.

We sincerely thank the reviewers and the AC for their time and thoughtful feedback on our paper. We hope that our responses have effectively addressed all questions and concerns, and we eagerly await further discussion and feedback.

---

### Decision · Program_Chairs · 2024-09-25

**Decision:**

Accept (poster)

**Comment:**

The paper "Make-An-Agent: A Generalizable Policy Network Generator with Behavior-Prompted Diffusion" introduces an approach for generating control policies using conditional diffusion models based on demonstrations of desired behaviors. The authors propose encoding behavior trajectories into embeddings, which are then used to generate latent policy parameter representations. These representations are subsequently decoded into functional policy networks. The reviewers acknowledge the originality of the idea, the good empirical performance, and the generalization ability of the proposed method.

The strengths of the paper lie in its use of conditional diffusion models for policy generation, which represents a departure from traditional policy learning methods. Make-An-Agent demonstrates the ability to generate effective policies for a wide range of tasks by conditioning on behavior embeddings, which shows its scalability across different domains. The diffusion-based generator exhibits strong generalization capabilities, producing proficient policies even for unseen behaviors and unfamiliar tasks. Moreover, the method can generate diverse and resilient policy parameters, maintaining high performance under environmental variability and with noisy input data. The effectiveness of Make-An-Agent is validated not only in simulations but also in real-world robotics, highlighting its practical applicability and efficiency.

The reviewers raised several important questions and concerns, which the authors addressed in their rebuttal. They provided additional experiments and analyses to support their claims, including the relationship between the effectiveness of generated policies and condition trajectories, the computational budgets compared with baselines, and generalization results on more diverse unseen tasks. The authors also included results of policy generation using expert demonstrations and compared them with offline RL baselines, demonstrating the superiority of their approach.

While one reviewer (Reviewer **YLrf**) recommended rejection, the concerns were not as in-depth as those raised by the other reviewers. The authors have earnestly responded to all the reviews, providing detailed explanations and additional experimental results to address the concerns. As the AC, I do not place significant weight on the reasons provided by Reviewer **YLrf** for rejection, as the authors have adequately responded to the concerns raised, and Reviewer **YLrf** did not respond further.

The reviewers' concerns have mostly been adequately responded to in the rebuttal, and the authors have committed to improving the paper based on the feedback received.  Based on the above reasons, the AC recommends accepting this paper for publication.